

# Diagnosing the decline in climatic mass balance of glaciers in Svalbard over 1957-2014.

Østby Torbjørn Ims[1], Schuler Thomas Vikhamar[1], Hagen Jon Ove[1], Hock Regine[2,3], Kohler Jack[4], and Reijmer Carleen H[5]

[1]Institute of Geoscience, University of Oslo, PO Box 1047 Blindern,N-0316 Oslo, Norway
[2]Geophysical Institute, University of Alaska, Fairbanks, Alaska 99775-7320, USA
[3]Department of Earth Sciences, Uppsala University, Villavägen 16, SE-75236 Uppsala, Sweden.
[4]Norwegian Polar Institute, Fram Centre, PO Box 6606 Langnes, N-9296 Tromsø, Norway
[5]Institute for Marine and Atmospheric Research, Utrecht University, Princetonplein 5, 3584 CC Utrecht, the Netherlands.

*Correspondence to:* (torbjorn.ostby@geo.uio.no)

**Abstract.** Longterm mass balance of all glaciers of the high Arctic Svalbard archipelago is difficult to achieve due to spatial and temporal incompleteness of geodetic and direct glaciological measurements. To close these gaps, we use a coupled surface energy balance and snow pack model to analyze Svalbard glacier mass changes and its evolution for the period 1957-2014. The model is forced by ERA-40 and ERA-Interim reanalysis data downscaled to 1 km resolution. Model validation is based on measured snow/ firn temperature and density, mass balance from stakes and ice cores, meteorological measurements, snow depths from radar profiles and remotely sensed surface albedo and skin temperatures. Overall model performance is good, but varies regionally.

Over the entire period the model yields a climatic mass balance of 8.2 cm w.e. yr$^{-1}$ which correspond to a mass surplus (excluding frontal ablation) of 175 Gt. Climatic mass balance has a linear trend of -1.4±0.4 cm w.e. yr$^{-2}$ with a shift from a positive to negative regime around 1980. Modeled mass balance exhibit large interannual variability, which is controlled by summer temperatures and further amplified by albedo feedback. For the period 2004-13 climatic mass balance was -21 cm w.e. yr$^{-1}$, and accounting for frontal ablation estimated by Błaszczyk et al. (2009) yields a total Svalbard mass balance of -39 cm w.e. yr$^{-1}$ for this 10 year period. In terms of eustatic sea level, this corresponds to a rise of 0.037 mm yr$^{-1}$.

Refreezing of water in snow and firn is substantial at 22 cm w.e. yr$^{-1}$, or 26 % of the accumulation. However, as warming lead to reduced firn area over the period, refreezing decrease both absolutely and relative to the mass budget. Negative mass balance and elevated equilibrium lines result in a massive loss of the thick firn (>2 m) extent and an increase of the superimposed ice, thin firn (<2 m) zone and bare glacier ice extent. Climate warming also causes a marked change in the thermal regime, with cooling of the glacier mid-elevation, and warming in the ablation zone and upper firn areas. By removing the thermal barrier, this warming has implications for the vertical transfer of surface meltwater through the glacier body and down to the base. In turn, this may influence basal hydrology, sliding and thereby overall glacier motion.





## 1 Introduction

Glaciers are widely acknowledged as good indicators of climate change (e.g. AMAP, 2011), but the relationship between atmosphere, surface energy balance and glacier mass balance is complex. Small glaciers and ice caps are currently among the major contributors to current sea level rise (Church et al., 2011), despite their relative small volume compared to the ice

sheets of Greenland and Antarctica and are assumed to be important throughout the 21st century (Meier et al., 2007). The high Arctic archipelago Svalbard has an estimated total eustatic sea level rise potential of 17-26 $mm$ (Martín-Español et al., 2015; Huss and Farinotti, 2012; Radić and Hock, 2010). Global glacier mass balance assessments suggest that Svalbard is one of the most important regional contributors to see level rise outside of Greenland and Antarctica over the 21st century (Giesen and Oerlemans, 2013; Marzeion et al., 2012; Radic et al., 2014), due to its location in one of the fastest warming

regions on Earth.

Through feedbacks in the climate system, the Arctic region experiences a greater warming than the global average, the so-called Arctic Amplification (e.g. Serreze and Francis, 2006). For the moderate emission scenario RCP4.5, Svalbard has a predicted warming of 5-8 $°C$ and a precipitation increase of 20-40 % by 2100 relative to the period 1986-2005 (IPCC, 2013). Since the 1960s, there has been a strong warming of $0.5\ °C\,decade^{-1}$ in Svalbard, the strongest warming measured in Eu-

rope (Nordli et al., 2014). Simultaneously there was a precipitation increase of $1.7\ \%\,decade^{-1}$ (Førland and Hanssen-Bauer, 2000). Steady negative glacier mass balance has been recorded since glaciological measurements began in 1967. However, direct measurements are mostly restricted to glaciers along the western coast of Svalbard, which is known to have more negative mass balance than the rest of the archipelago (Hagen et al., 2003a). Climatic mass balance ($B_{\mathrm{clim}}$), the sum of surface mass balance and internal accumulation (Cogley et al., 2011), as derived from ice cores over 1960-2000 (Pinglot et al., 1999, 2001;

Pohjola et al., 2002) and modeled balances for the period 1979-2013 (Lang et al., 2015a) shows no trends. In contrast, geodetic mass balance studies indicate accelerated glacier mass loss over the last decades (James et al., 2012; Kohler et al., 2007; Nuth et al., 2007). There are multiple causes for this apparent disagreement. Geodetic approaches include all components of the mass budget, i.e. the climatic balance and mass losses through calving and submarine melting at tidewater glacier termini. In addition, ice cores are taken in the accumulation area, while trends in the ablation area may differ; the latter have been

shown to have a substantial effect on the glacier wide climatic balances in Svalbard (e.g. Aas et al., 2016; van Pelt et al., 2012). Meteorologically driven mass balance models facilitates filling of these spatial and temporal gaps, however care must be taken to adequately represent spatial and temporal scales of relevant processes. For instance, the coarse spatial 10 $km$ grid used by Lang et al. (2015a) does not represent the glacier hypsometry at lower elevations thereby influencing the results.

Here we present results from a model study that covers the entire archipelago for the period $1957-2014$ at unprecedented

spatial and temporal resolution. At time steps of 6 hours, we calculate the mass and energy fluxes at the glacier surface and in the subsurface layers using DEBAM (Distributed Energy Balance Model) developed by Hock and Holmgren (2005) and Reijmer and Hock (2008). ERA-40 and ERA-Interim climate reanalysis data are downscaled to 1 $km$ horizontal resolution largely following the TopoSCALE methodology (Fiddes and Gruber, 2014), except for precipitation, where we use the Linear theory for orographic lifting (Smith and Barstad, 2004). We do a thorough comparison with a large number and different types





of observations to validate model performance. From our model results we identify climatic controls on the climatic mass balance over the study period and discuss implications for the future by testing sensitivities and simulations for a 2100 climate as predicted by Førland et al. (2011). We also examine modeled responses of the water retention capacity in a warmer climate and discuss related implications for $B_{\mathrm{clim}}$ and ice dynamics through changes in the hydrological and thermal regimes.

## 2 Svalbard climate and target glaciers

The Svalbard archipelago is located in the Norwegian Arctic between 75-81 °N (Fig. 1). The land area of the islands is ∼60 000 km² of which 57 % are covered by glaciers (Nuth et al., 2013). While the western side of the archipelago is characterized by alpine topography the eastern side has less rugged topography and many low altitude ice caps. Through the Norwegian current, an extension of the Gulf stream, warm Atlantic water is advected northwards to keep the western side of Svalbard mostly ice-free year-round (Walczowski and Piechura, 2011). In contrast, the ocean east of Svalbard is dominated by Arctic ocean currents (Loeng, 1991). Similarly contrasting regimes are found in the atmosphere, where warm and moist air is associated with southerly flow while colder and drier air masses originate from the northeast (Kaesmacher and Schneider, 2011). These oceanic and atmospheric circulation patterns combined with the fluctuating sea ice edge cause large temporal and spatial gradients of temperature and precipitation across the archipelago (Hisdal, 1998). Therefore, Svalbard has been identified as one of the most climatically sensitive regions of the world (Rogers et al., 2005).

The climate of Svalbard is polar maritime, where both rain and snowfall may occur in all months of the year. At the main settlement Longyearbyen, mean annual air temperature for the normal period 1961-1990 is -6.7 °C. A warming trend of 2.6 °C century⁻¹ has been identified from the 117 year long Svalbard Airport temperature record (Nordli et al., 2014), see also Supplement (Fig. S1-S2). Although a positive trend exists for all seasons, the annual trend is dominated by an increase of winter temperatures. Increased air temperatures and precipitation has occurred simultaneously with reduced sea ice ice cover around Svalbard (Rodrigues, 2008).

Annual precipitation in Longyearbyen is 190 mm and has increased by 2.5 % decade⁻¹ over the last 80 years (Førland et al., 1997; Hanssen-Bauer and Førland, 1998), although precipitation gauge undercatch hampers trend analysis (Førland et al., 1997; Hanssen-Bauer and Førland, 1998; Førland and Hanssen-Bauer, 2003). Large precipitation variability is observed across the archipelago, with about three times higher precipitation along the west coast compared to Longyearbyen (Førland and Hanssen-Bauer, 2003) and even more in the southern Spitsbergen (Sand et al., 2003; Winther et al., 2003). The drier central Spitsbergen has less extensive glacier coverage and is characterized by land terminating cirque and valley glaciers.

Mass balance is measured at stakes along the center profiles of five glaciers in (see Fig. 1). Etonbreen with its 640 km² is the largest of the observed glaciers and drains gently westwards from the summit of the Austfonna ice cap. Measured surface mass balance is almost in balance while mass loss is mainly caused by margin retreat, which has been retreated since the last surge in the 1930s. Kongsvegen (173 km²) and Holtedahlfonna (385 km²) are situated in northwest Spitsbergen. The ice field Holtedahlfonna feeds Kronebreen, a steady fast-flowing glacier, while Kongsvegen is nearly stagnant since it is in the quiescent surge phase (Kääb et al., 2005). Kongsvegen and Kronebreen merge downstream and enter Kongsfjorden together.



Despite the short distance, Kongsvegen has higher accumulation rates than Holtedahlfonna, but because of their hypsometry, surface mass balance on Kongsvegen (-4 cm w.e. yr$^{-1}$) is slightly more negative than Holtedahlfonna (-2 cm w.e. yr$^{-1}$) over 1966-2007 (Nuth et al., 2012). Nordenskiöldbreen (200 km$^2$) is a valley glacier located in the central Spitsbergen flowing southwest from the Lomonosovfonna ice field entering the Adolfbukta fjord. Mass balance has been measured since 2006 and is negative (van Pelt et al., 2012). Hansbreen (56 km$^2$) is a calving glacier where mass balance has been measured since 1989. Redistribution by wind is found to be important on Hansbreen and the mass balance has been negative by -28 cm w.e. yr$^{-1}$ since 1989 (Grabiec et al., 2006, 2012).

## 3  Data

### 3.1  Topography and glacier masks

The 1 km resolution digital elevation model (DEM), used in this study, was resampled from a 90 m DEM after Nuth et al. (2007). Slope and aspect was computed following Zevenbergen and Thorne (1987). Fractional glacier masks were created by computing the percentage of glacier coverage for each grid point of the 1 km DEM for three periods (1930-60s, 1990s, and 2000s) based on the multi-temporal inventory by Nuth et al. (2013). However, only the latest DEM has a complete coverage over all of Svalbard (see Fig. S3). We created a fourth fractional glacier mask that combines the masks from the three periods so that each grid cell contains the largest glacier extent of any of these periods (henceforth referred to as the reference glacier mask). Finally, an annual time series of glacier masks was created by linear interpolation thus assuming linear glacier retreat or advance between the epochs and no changes after the 2000s epoch (see Fig. S4).

Small discrepancies are introduced by converting glacier polygons to the 1 km grid. Whereas the glacier cover in the 2000s inventory is 33775 km$^2$ the fractional glacier mask is 14 km$^2$ (0.04%) larger. The reference glacier mask covers 36943 km$^2$ and is 10 % larger than the 2000s inventory. Figure 2 shows the hypsometry of each region and all regions combined covers the entire Svalbard glacier area. The error (difference between 90 m and 1 km DEM) is generally low, but glacier area in the 1 km DEM is slightly underestimated below 150 m a.s.l.

### 3.2  Downscaled ERA-40/ Interim climate reanalysis data

The glacier model is forced by fields of downscaled near-surface air temperature, relative humidity, wind and downwelling shortwave and longwave radiation from the ERA-40 and Interim reanalyses of the European Centre for Medium-Range Weather Forecasts (Uppala et al., 2005; Dee et al., 2011). The reanalysis data of 6-hourly temporal resolution is provided on a 0.75 °×0.75 °spatial grid which is shown in Figure 3, and covers the periods 1957-1979 (ERA-40) and 1979-2014 (ERA-Interim).

Precipitation is often heavily biased in coarsely-resolved reanalyses, especially in environments with pronounced topography where it typically is too low and lacks spatial detail (Schuler et al., 2008). This is associated with the smoothed representation of the actual topography in the large-scale model used for the reanalysis (Fig. 3), leading to an underestimate of orographic precipitation enhancement. We assume that this is the main reason for the poor performance of reanalyzed precipitation and use





**Figure 1.** Overview map of Svalbard with names of regions and the five glaciers described in the text, glacierized area shown in blue. Markers show the location of observations and the black dots mark the calibration sites. Lines mark elevation above sea level in 200 m intervals.

instead a linear theory (LT) of orographic precipitation (Smith and Barstad, 2004) to downscale ERA-precipitation to our 1 km resolution model domain. The LT-model describes the motion of an air parcel, characterized by its temperature, stability, wind





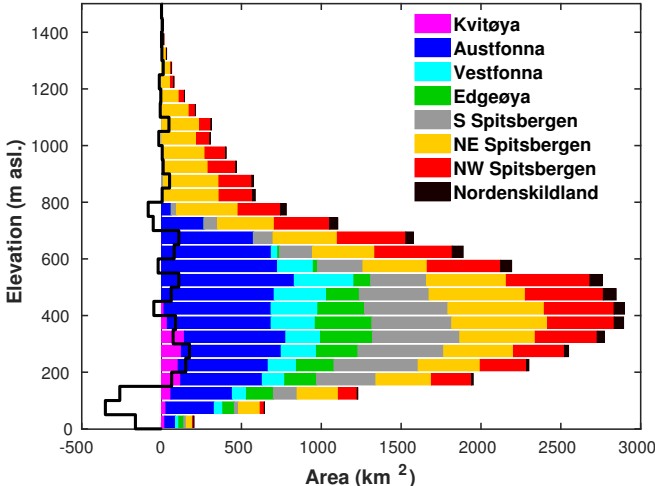

**Figure 2.** Regional glacier hypsometry of the 2000s inventory. Stacked bars correspond to the entire Svalbard glacier area. Altitude intervals of 50 m. The black line is the difference between hypsometry of the 90 m DEM and the applied 1 km DEM, for the entire Svalbard.

direction and speed. Terrain induced uplift of the air parcel results in condensation and eventually precipitation of moisture further downstream of the uplift. This model has been successfully evaluated using precipitation gauges (Barstad and Smith, 2005) and snow measurements (Schuler et al., 2008) and applied for downscaling precipitation (e.g. Crochet et al., 2007). To discriminate solid from liquid precipitation, a simple thresholding approach was used, assuming that all precipitation is liquid

at temperatures above 2.5 °C, all is solid at temperatures below 0.5 °C, with a linear transition between.

The other required climate variables are downscaled to the 1 km grid using the TopoSCALE methodology (Fiddes and Gruber, 2014). TopoSCALE exploits the relatively high vertical resolution of the reanalysis data, and downscaled variables at the actual topography are based on the properties of the atmospheric column in the reanalysis. The downscaled fields preserve the horizontal gradients present in ERA, but include additional features caused by the real topography not present in ERA (Fig. 4).

This approach is assumed to outperform simpler bias corrections since transient properties of the atmospheric column are used. For example any inversions in the reanalysis data will be preserved in the downscaled product.

We modify the TopoSCALE methodology regarding downscaling of direct shortwave radiation and air temperature. For direct solar radiation we apply the relationship after Kumar et al. (1997) for atmospheric attenuation rather than the one given in Fiddes and Gruber (2014). Solar geometry variables such as solar zenith and azimuth, and self-shading due to local slope and

aspect are calculated following Reda and Andreas (2004). Cast shadow and hemispherical obstructions caused by surrounding topography are calculated following Ratti (2001).

During summer when air temperatures ($T_{\mathrm{air}}$) are above freezing, the TopoSCALE method resulted in too high $T_{\mathrm{air}}$ wherever the actual topography is above the ERA topography and too low temperatures where the actual topography is situated below. These anomalous temperatures resulted from extrapolation of the typical near-surface inversion above a melting glacier surface.

Such strong near-surface inversions occur since surface temperatures are restricted to the melting point in contrast to air



temperatures which may be warmer. Hence, the extracted lapse rate indicates inversion while this is not true for the free atmosphere. To avoid extrapolation using the anomalous lapse rates we employ the ERA 2 m $T_{\mathrm{air}}$ in the downscaling under these conditions. This implies that negative lapse rates (inversion) are set to zero whenever melt occurred in the reanalysis.

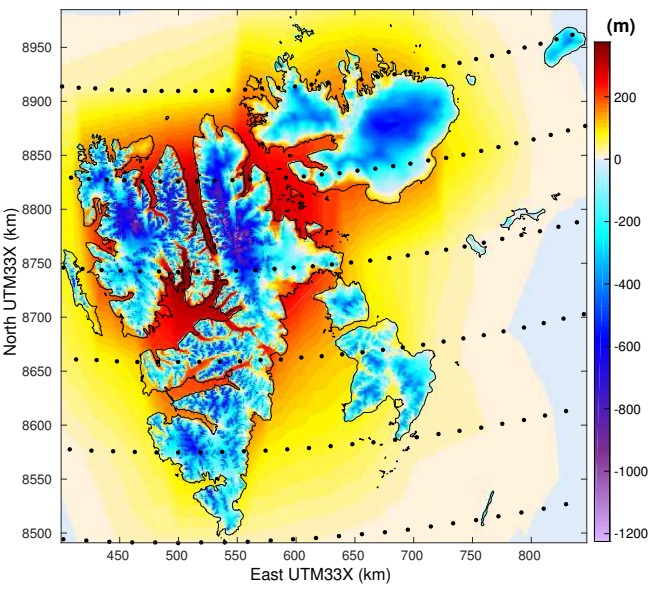

**Figure 3.** Difference (m) between the ERA topography and the 1 km DEM; negative values mean that ERA elevations are lower. Black dots show the 0.75 $^\circ$ ×0.75 $^\circ$ ERA grid.

### 3.2.1 Validation of climate input

Downscaled variables are compared to observations at the meteorological stations listed in Table 1 mostly for the period after 2004. Daily averages of air temperature, relative humidity, wind speed and radiation and monthly precipitation of the closest model grid point are compared to the corresponding variable at the measuring site. Air temperatures (2 m) is by far the most commonly measured variable, while radiation components are available only at two stations (Table 1). Despite altitude differences of up to 100 m between measuring site and corresponding cell in the model, no altitude correction is performed

due to unknown lapse rates.

In general there is a good agreement between downscaled ERA and observed air temperatures with biases mostly below 1.5 K (Tab. 2). Despite small bias averaged over the year, there is a clear seasonal bias, where ERA temperatures are too warm during winter and too cold during summer (Fig. 5). Although the biases in Figure 5 are negative during summer, ERA is too warm over the glaciers during summer when 2 m air temperatures are above freezing.

At Svalbard Airport, the performance of downscaled ERA-40 and ERA-Interim are investigated for the entire model period. Over 1957-1979 only monthly measured temperatures are available at Svalbard Airport, where downscaled ERA-40 has a monthly RMSE of 2.3 °C. For the 1979-2002 period the reanalysis products overlap with monthly RMSE of 1.8 °C and 1.5 °C





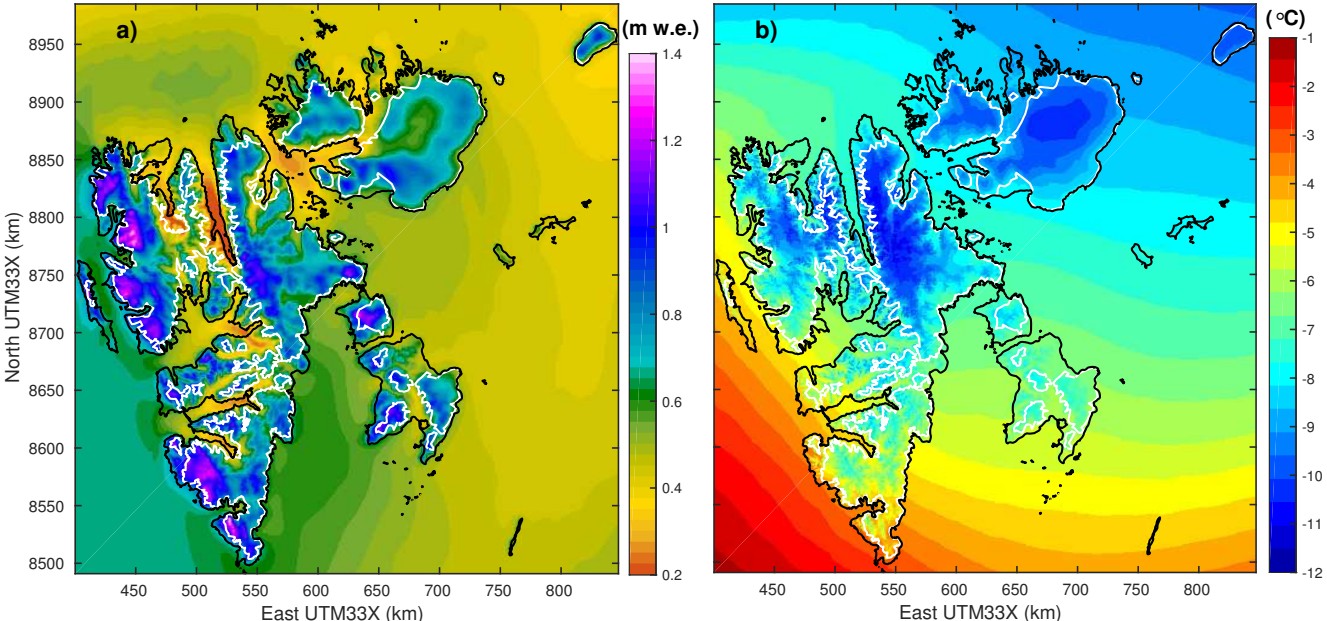

**Figure 4.** Downscaled Precipitation (**a**) and 2 m air temperature (**b**) as annual mean over the ERA-Interim period (1979-2014).

at Svalbard Airport for ERA-40 and ERA-Interim, respectively. We attribute the lower performance prior to 1979 to the lack of satellite observations for constrain sea surface temperatures and sea ice cover in the reanalysis. Since the Svalbard Airport temperature record and other sites at the west coast likely are incorporated into the reanalysis, the quality of the reanalysis in the pre-satellite era is possibly even lower in remote areas with no observations. The annual observed air temperature trend for the period 1957-2013 at Svalbard Airport is $0.70 \pm 0.22\,^{\circ}\mathrm{C}\,\mathrm{decade}^{-1}$, while the downscaled ERA data has an insignificantly lower warming trend of $0.67 \pm 0.19\,^{\circ}\mathrm{C}\,\mathrm{decade}^{-1}$ at Svalbard Airport.

Downwelling shortwave and longwave radiation are compared to measurements at Etonbreen (Schuler et al., 2014) and the Baseline Surface Radiation Network site (BSRN) in Ny-Ålesund (Maturilli et al., 2013). In Ny-Ålesund the model largely reproduces observations both for short and longwave radiation. During winter, downwelling longwave radiation is slightly underestimated while there is no bias during summer. Since there is no temperature bias in Ny-Ålesund during winter, the underestimation of longwave radiation is indicative of a too thin cloud cover in the reanalysis. Representations of clouds are among the major issues of the reanalysis (Aas et al., 2016). Downwelling shortwave radiation is overestimated by $7\,\mathrm{W\,m}^{-2}$ over the summer season in Ny-Ålesund. There is a much better agreement with radiation observations in Ny-Ålesund than in northeastern Svalbard. This is to be expected, since radio soundings and other observation data from Ny-Ålesund are assimilated into ERA-Interim. Therefore, clouds in the reanalysis are much better represented in Ny-Ålesund than at Austfonna.

On Etonbreen during summer, downwelling shortwave radiation is underestimated by $40\,\mathrm{W\,m}^{-2}$ while downwelling longwave radiation is overestimated by $12\,\mathrm{W\,m}^{-2}$, indicative of a too thick atmosphere or too many clouds in the reanalysis. However, these biases could be partly explained by measurement uncertainty caused by rime on the sensor or by sensor tilt.



**Table 1.** Meteorological stations used for validation of the downscaled ERA-Interim reanalysis. $N$ indicates the number of daily averages used in the validation, subscript "T" refers to 2 m air temperature, "RH" is relative humidity, "ws" is wind speed, "rad" is downwelling short and long wave radiation and "prec" is precipitation. Also given is the elevation of the observation site ($Z$) and the closest grid point ($Z_{\mathrm{DEM}}$).

| Location | Long | Lat | $Z$ (m asl.) | $Z_{\mathrm{DEM}}$ (m asl.) | Period | $N_{\mathrm{T}}$ | $N_{\mathrm{RH}}$ | $N_{\mathrm{ws}}$ | $N_{\mathrm{rad}}$ | $N_{\mathrm{prec}}$ |
|---|---|---|---|---|---|---|---|---|---|---|
| Etonbreen[†] | 22.42 | 79.73 | 369 | 350 | 2004 –d.d. | 3295 | 2738 | 2913 | 3240 | 0 |
| Janssonhaugen | 16.47 | 78.18 | 270 | 163 | 2011 –d.d. | 910 | 0 | 945 | 0 | 0 |
| Gruvefjellet | 15.62 | 78.20 | 464 | 359 | 2007 –d.d. | 2555 | 2555 | 2551 | 0 | 0 |
| Kapp Heuglin | 22.82 | 78.25 | 18 | 4 | 2006 –d.d. | 2099 | 0 | 2112 | 0 | 0 |
| Rijpfjorden | 22.48 | 80.22 | 10 | 31 | 2007 –d.d. | 1495 | 1495 | 1304 | 0 | 0 |
| Svalbard Airport | 15.47 | 78.25 | 28 | 3 | 1976 –d.d. | 12777 | 0 | 12724 | 0 | 199* |
| Isfjord Radio | 13.63 | 78.07 | 13 | 1 | 2000 –2006 | 1666 | 0 | 0 | 0 | 0 |
| Verlegenhuken | 16.25 | 80.06 | 8 | 0 | 2011 –d.d. | 986 | 0 | 1700 | 0 | 0 |
| Hornsund | 15.54 | 77.00 | 10 | 8 | 1996 –d.d. | 4635 | 0 | 4473 | 0 | 95* |
| Kvitøya | 31.50 | 80.07 | 10 | 17 | 2012 –d.d. | 740 | 0 | 702 | 0 | 0 |
| Holtedahlfonna[†] | 13.62 | 78.98 | 688 | 702 | 2009 –2010 | 317 | 0 | 265 | 0 | 0 |
| Ny-Ålesund | 11.93 | 78.92 | 8 | 6 | 1975 –d.d. | 12666 | 12708 | 12349 | 0 | 212* |
| Bayelva | 11.83 | 78.92 | 25 | 28 | 2003 –d.d. | 3652 | 0 | 0 | 3652 | 0 |
| Hopen | 25.01 | 76.51 | 6 | 60 | 1957 –d.d. | 13178 | 0 | 13044 | 0 | 306* |

*: Number of months

[†]:Station located on glacier.

The latter issue is caused by the fact that the ice foundation of an autonomous weather stations may melt and deform causing tilt and thereby large errors, especially at high solar zenith angles (Bogren et al., 2015). The latter issue is caused by melt and deformation of the foundation for the autonomous weather stations causing sensor tilt, changed hemispherical view and thereby large errors, especially at high solar zenith angles (Bogren et al., 2015).

5    Wind speeds are reproduced reasonably well including the seasonal cycle. Biases are within $\pm 1.5\ \mathrm{m\,s}^{-1}$ with no clear seasonal trend. It is likely that the biases are caused by site specific effects, such as deceleration of air flow in the lee of a topographic obstacle or acceleration due to channelizing through valleys.

For relative humidity the reanalysis represents the seasonality well, and in late summer both the humidity and the biases are of largest magnitude. At the two coastal stations, the downscaled reanalysis is too dry whereas it is too humid at the higher

10    elevated stations. The coarse land mask of the reanalysis and the poor representation of sea ice are most likely the main causes for these biases.

Downscaled precipitation is overestimated by 5 to 25 mm at the weather stations, with a slightly higher bias during winter. These biases are partly caused by measurement undercatch, which is reported to be 50 % at Svalbard on an annual basis, with higher undercatch during winter than during summer (Førland and Hanssen-Bauer, 2000).





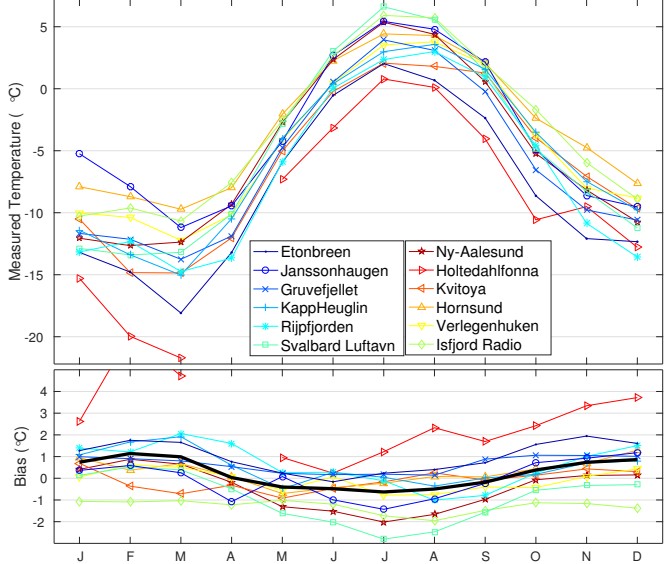

**Figure 5.** Monthly observed air temperatures (upper panel) and monthly biases (lower panel). Bold black line is average over the twelve sites.

## 3.3 Satellite-derived surface temperatures and albedo

### 3.3.1 MODIS skin surface temperatures

Skin surface temperatures ($T_{surf}$) derived from the Moderate Resolution Imaging Spectroradiometer instrument (MODIS) are used to constrain modeled surface temperatures. We employ the MODIS level 3 collection 5 products MOD11A1 (Terra space craft) and MYD11A1 (Aqua space craft) where surface temperatures are retrieved by the Split-window algorithm (Wan and Dozier, 1996; Wan, 2008). Cloudy satellite scenes are masked out using the MODIS cloud mask products M(O/Y)D35_L2 (Ackerman et al., 1998; Frey et al., 2008). Under cloud-free conditions Terra and Aqua combined provides four estimates of $T_{surf}$ per day at 1000 m resolution. Automatically generated quality control flags represent the confidence level of the produced $T_{surf}$. As suggested by Østby et al. (2014), observations flagged as "other quality" and "$T_{surf}$ error<3" have been excluded. MODIS $T_{surf}$ has usually an accuracy better than 1 K (Wan, 2014). However, a much lower accuracy is found for snow and ice surfaces due to the ambiguous cloud detection caused by the spectral similarities of snow and clouds (Hall et al., 2004, 2008; Scambos et al., 2006; Østby et al., 2014). For Svalbard Østby et al. (2014) found a RMSE=5.0 K and bias of -3.0 K. Figure S5 in the supplement show average $T_{surf}$ and minimum albedo from MODIS.

### 3.3.2 MODIS albedo

Similar to the MODIS surface temperatures, daily satellite-derived albedo is provided by the snow cover products M(O/Y)D10A1 at 500 m spatial resolution (Hall and Riggs, 2007). To minimize possible errors in the satellite albedo, acquisition during days



**Table 2.** Seasonal biases (modeled minus observational averages) at all observation sites averaged over each site's observation period (Table 1). Column heading of $S$ and $W$ denotes summer (Jun-Aug) and winter (Sep-May), respectively. Positive numbers indicate that the model overestimate observations.

| Location | $T_{air}$ (°C) | | RH (%) | | WS ($m\,s^{-1}$) | | $S_{\downarrow}$ ($W\,m^{-2}$) | | $L_{\downarrow}$ ($W\,m^{-2}$) | | Precip. (mm) | |
|---|---|---|---|---|---|---|---|---|---|---|---|---|
| | S | W | S | W | S | W | S | W | S | W | S | W |
| Etonbreen | 0.16 | 1.28 | -2.20 | -5.44 | 0.31 | 0.18 | -39.74 | -10.23 | 12.13 | -14.29 | – | – |
| Janssonhaugen | -1.12 | 0.31 | – | – | -1.81 | -1.30 | – | – | – | – | – | – |
| Gruvefjellet | 0.17 | 0.83 | 3.11 | -2.93 | 0.03 | -0.19 | – | – | – | – | – | – |
| Kapp Heuglin | -0.01 | 0.72 | – | – | -0.05 | 0.96 | – | – | – | – | – | – |
| Rijpfjorden | -0.28 | 0.94 | 5.66 | 2.59 | 0.83 | 0.93 | – | – | – | – | – | – |
| Svalbard Lufthavn | -2.43 | -0.42 | – | – | -1.32 | -1.03 | – | – | – | – | 24.6 | 24.9 |
| Isfjord Radio | -1.63 | -1.18 | – | – | – | – | – | – | – | – | – | – |
| Verlegenhuken | -0.49 | 0.04 | – | – | -1.85 | -1.70 | – | – | – | – | – | – |
| Hornsund | -0.19 | 0.43 | – | – | -0.03 | 0.40 | – | – | – | – | 5.4 | 18.7 |
| Kvitoya | -0.13 | -0.09 | – | – | -0.89 | -1.27 | – | – | – | – | – | – |
| Holtedahlfonna | 1.26 | – | – | – | -0.66 | – | – | – | – | – | – | – |
| Ny-Ålesund | -1.73 | -0.04 | 7.63 | 3.19 | 0.73 | 0.77 | – | – | – | – | 23.4 | 18.3 |
| Hopen | -1.87 | -5.19 | – | – | -0.32 | 0.11 | – | – | – | – | 7.4 | 10.1 |
| Bayelva | -2.13 | 0.01 | – | – | – | – | 6.89 | -3.50 | -1.60 | -8.84 | – | – |

of noon solar zenith above 70 ° were excluded (Schaaf et al., 2011). Mean daily albedo was only calculated if both Terra and Aqua had observation flagged "good" by the internal quality check. Observations were also discarded if the albedo difference between Aqua and Terra exceeded 0.1. Since such rapid and large albedo change only can occur during snowfall, when clouds anyway should preclude acquisition since clouds are opaque in the visible and thermal parts of the spectrum. Given these criteria, the satellite-derived albedo yielded RMSE=0.08, mean bias=-0.005 and $R^2$=0.72 compared to the noon time albedo measured at Etonbreen.

## 4 Model description

A surface energy balance model coupled to a snow pack model was used to calculate surface energy fluxes, mass balance, water retention and snow and ice properties. Only the main features will be described here. For model details see Reijmer and Hock (2008); Hock and Holmgren (2005); Østby et al. (2013). Our model set-up employs slightly different parameterizations for albedo, thermal conductivity and runoff than the original model.

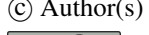



## 4.1 Surface energy balance

The energy balance at the glacier surface is given by:

$$Q_N + Q_H + Q_L + Q_R + Q_G + Q_M = 0, \tag{1}$$

where net radiation is $Q_N = S_\downarrow(1-\alpha) + L_\downarrow + L_\uparrow$. Downwelling short and longwave radiation fluxes are taken from down-
scaled ERA reanalysis data. Turbulent fluxes of sensible heat ($Q_H$) and latent heat ($Q_L$) are calculated from the Monin-
Obukhov-theory using downscaled humidity, wind speed and air temperatures at screen level (2 m). Roughness lengths of
momentum for snow and ice surfaces are determined through calibration (Sec. 4.4), while roughness lengths for heat and vapor
are calculated according to Andreas (1987). Sensible heat supplied by rain water ($Q_R$) is derived from the rainfall rate assum-
ing that the hydrometeors have the same temperature as the surrounding air. $Q_G$ is the energy exchange with the subsurface
layer (Sec. 4.2). Any energy surplus $Q_M$ is used for melting snow and ice. The sign convention is such that fluxes directed
towards the surface carry a positive sign and vice versa.

### 4.1.1 Albedo

Since shortwave radiation is generally the most important energy supply for melt on Arctic glaciers (e.g. Arendt, 1999;
Arnold et al., 2006; Østby et al., 2013), an albedo parameterization was carefully constructed, but quite similar to the one
of Bougamont and Bamber (2005). Albedo is set to a maximum value during snowfall. Snow aging induces exponential albedo
decay with different temperature dependent time scales for wet and dry snow. In case of a thin snow cover, albedo is re-
duced from snow albedo to the underlying albedo (of firn, ice or super imposed ice) using the relationship described by
Oerlemans and Knap (1998), with a characteristic snow depth scale of 3 cm, such that the albedo transition is smooth when
snow cover is thin. This behavior is essential in the Arctic, where precipitation events are frequent (~200 days a year), but
usually yield low amounts (Aleksandrov et al., 2005). A similar approach was used for albedo reduction to account for water
ponding at the surface with a characteristic water depth of 30 cm (Zuo and Oerlemans, 1996). Threshold values for albedo
of firn and ice are determined during calibration (Sec. 4.4). Albedo is adjusted for specular reflection at high zenith angles
following (Gardner and Sharp, 2010).

## 4.2 Subsurface processes

The subsurface model is based on the SOMARS mode (Greuell and Konzelmann, 1994) which calculates temperature, density
and water content of the subsurface layers and the subsurface energy flux $Q_G$. The surface energy balance and subsurface
model is connected through the skin surface temperature assuming the surface to be an infinite layer with zero heat capacity.
Percolation follows a tipping-bucket scheme, where water percolating downwards is partly retained by capillary forces or
refreezes upon encountering layers at sub-zero temperatures. Where water meets impermeable ice, slush builds up and lateral
runoff is computed using a relationships defined by Zuo and Oerlemans (1996). A density dependent thermal conductivity



parameterization (Douville et al., 1995) was calibrated for computing subsurface heat conduction. Irreducible water content is calculated after Schneider and Jansson (2004) and densification of dry snow after Herron and Langway (1980).

## 4.3 Climatic mass balance

The climatic mass balance is the sum of: melt and (re-)sublimation at the surface, refreezing in the subsurface layers and solid
precipitation from the downscaled reanalysis. Although the model also calculates the water balance, liquid water retained in snow or firn is not included in the mass balance (Cogley et al., 2011).

## 4.4 Model set-up and calibration

The model is run for each glacierized grid cell based on the reference glacier mask (Section 3.1). The temporal resolution of the surface energy balance is that of the ERA reanalysis (6 hours), while the snow model uses an internal time step of 3 minutes for
which the ERA forcing is linearly interpolated. The subsurface model is solved on an adaptive grid consisting of 15-35 layers with maximum depth of ∼40 m below the surface. Layers close to the surface are few cm thick, while the layers closer to the bottom are several m thick.

Snow/ice temperature, density and water content are initialized with a 10 year spin-up using the climate data of the 1960s. To start the spin-up, the entire subsurface domain has an initial density of ice ($900 \, \mathrm{kg \, m^{-3}}$), zero water content and temperatures
of 0 °C at the surface and linearly decreasing to -5 °C at 7 meters depth, beyond which the temperature stays constant. The initial -5 °C at depth is somewhat higher than the mean annual air temperature and is based on observed temperatures in boreholes (Bjørnsson et al., 1996). Different initial conditions for the spin-up period had little effect on the results at most locations where the spin-up time was tested. A few exceptions occurred at some locations close to the equilibrium line, where the system memory seems to be longer. At these locations differences in annual mass balance of up to $5 \, \mathrm{cm \, w.e. \, yr^{-1}}$ occurred
in the first years following the spin-up period, when a 10 year spin-up is run twice instead of once.

Since model calibration is computationally expensive, only 48 grid cells were selected for calibration. Most of these sites correspond to locations where mass balance is monitored, but additional sites were included to achieve a good spatial coverage over the entire archipelago (Fig. 1) At these additional sites only remote sensing data were available for calibration. Glacier edges were avoided such that the MODIS data (1 km & 0.5 km resolution) were entirely on glacier when resampled to the
applied 1 km DEM.

Even though the model is physically based, some parameters do not have a consensus estimate in the literature. Østby et al. (2014) showed that model performance is sensitive to the choice of parameters concerning the turbulent fluxes (roughness lengths) and the albedo parameterization. Here, the model is calibrated using an adaptive Monte Carlo Markov chain solver entitled DREAM$_{\mathrm{ZS}}$ (Differential Evolution Adaptive Metropolis) (Vrugt et al., 2009; Laloy and Vrugt, 2012). This approach
enables parallel computing, since the Markov chains evolve almost separately. Four parameters; roughness lengths for snow and ice surfaces, and albedo for firn and ice were selected for calibration. DREAM$_{\mathrm{ZS}}$ seeks to maximize the likelihood $L$ of an





**Table 3.** Data used for calibration: Sites indicate how many of the 48 sites where the respective quantities are available, $N$ is total number of observations for each quantity at all sites, and the observation uncertainty ($\sigma$). The indices $a$, $s$ and $w$ for measured surface mass balance $b$ are annual, summer and winter averages, respectively. "SR" from the Automatic weather station on Etonbreen is surface height changes from a sonic ranger.

| Data (unit) | Sites | $N$ | $\sigma$ |
|---|---|---|---|
| $b_a$ ( cm w.e.) | 30 | 268 | 10 |
| $b_s$ ( cm w.e.) | 30 | 259 | 10 |
| $b_w$ ( cm w.e.) | 30 | 294 | 10 |
| MODIS $\alpha$ (%) | 48 | 9386 | 8 |
| MODIS $T_{\mathrm{surf}}$ (K) | 48 | 315615 | 5 |
| AWS $\alpha$ (%) | 1 | 1424 | 4 |
| AWS $L_\uparrow$ ( W m$^{-2}$) | 1 | 10826 | 3 |
| AWS SR (cm) | 1 | 10384 | 3 |
| AWS $T_{ice}$ (K) | 1 | 33222 | 0.25 |

observed quantity ($O$) and its corresponding quantity predicted by the model ($P$) using

$$L = \frac{-N \ln(2\pi)}{2} - \sum_i^N \ln(\sigma_i) - \frac{\left( \sum_i^N \left( \frac{P_i - O_i}{\sigma} \right)^2 \right)}{2} \tag{2}$$

where N is the number of samples and $\sigma$ is the uncertainty of the measurement.

Nine different types of data are used for calibration. Table 3 lists the variables along with the measurement uncertainty $\sigma$
5 and number of observations $N$. Uncertainty of stake mass balance is assumed to be 10 cm w.e., in the lower range of reported uncertainties (e.g. Huss et al., 2009; Zemp et al., 2013). We adopt this low value due to the low mass turnover in Svalbard, but we acknowledge that higher uncertainties are likely for the accumulation area. For MODIS-derived albedo we apply an uncertainty of 8 % (Sec.3) and 5 K for MODIS $T_{\mathrm{surf}}$ (Østby et al., 2014). For the measurements at the automatic weather station we assume $\sigma$=3 W m$^{-2}$ for longwave radiation (Michel et al., 2008), while uncertainties for the albedo, the sonic
10 ranger data and snow and ice temperatures from the thermistor string, are given by the instrument manufactures (Schuler et al. (2014); Østby et al. (2013)).

Likelihood functions for each of the nine data types are summed and the parameter set yielding the highest total $L$ is applied in the simulations. Calibrated parameter values and other model parameters are listed in Table S1.





## 5 Results

### 5.1 Climatic mass balance

The modeled mean annual $B_{\mathrm{clim}}$ averaged over the entire domain for the period 1957 to 2014 is positive (8.2 cm w.e. $\mathrm{yr}^{-1}$), which corresponds to a mass gain of 3.1 $\mathrm{Gt\,yr}^{-1}$ using the current glacier mask of each year. However, there is considerable temporal and spatial variability (Fig. 6). We find glacier-wide mass loss for many southern Spitsbergen glaciers. In contrast, northern Spitsbergen and Nordaustlandet have positive $B_{\mathrm{clim}}$ and in some cases the accumulation area almost reaches sea level. The components of the climatic mass budget averaged over all glaciers and the 1957 - 2014 simulation period are shown in Table 4. Ablation is dominated by melt (98 %) while refreezing is a major component (26 %) of accumulation.

The temporal variations of the glacier-wide annual balances and their components are shown in Figure 7. Despite overall positive $B_{\mathrm{clim}}$, there is a clear negative trend of 14±4.1 cm w.e. $\mathrm{decade}^{-1}$ over the entire period 1957-2014. Since we identified some weakness in the ERA40 data (see Sec 6.3) we extract the $B_{\mathrm{clim}}$ trend for the ERA-interim period 1979-2014, which yields a trend of -9.6±9.9 cm w.e. $\mathrm{decade}^{-1}$, thus not significant at the 95 %-level. Melt is well correlated ($R^2 = 0.93$) with $B_{\mathrm{clim}}$, and controls both $B_{\mathrm{clim}}$ interannual variability and the trend. Accumulation has a small increase (not significant at 95 %-level) over the period while there is a significant decrease in refreezing.

**Table 4.** Climatic mass balance components in cm w.e. $\mathrm{yr}^{-1}$ averaged over all glaciers in Svalbard for the period 1957-2014 and standard deviation (STD) of the temporal variability, temporal correlation with $B_{\mathrm{clim}}$ ($R_{B_{\mathrm{clim}}}$), trend slope ($\beta$) in cm w.e. $\mathrm{yr}^{-2}$ and slope uncertainty ($2\sigma_{slope}$). Slope significance at the 95 %-level in bold font ($|\beta| > 2\sigma_{slope}$).

| Variable | Mean | STD | $R_{B_{\mathrm{clim}}}$ | $\beta$ | $2\sigma_{\mathrm{slope}}$ |
|---|---|---|---|---|---|
| $B_{\mathrm{clim}}$ | 8.2 | 34.2 | 1.00 | **-1.35** | 0.41 |
| Snowfall | 61.3 | 9.3 | 0.32 | 0.10 | 0.14 |
| Rime | 1.1 | 0.3 | -0.38 | **0.01** | 0.00 |
| Refreezing | 21.5 | 3.7 | 0.62 | **-0.12** | 0.05 |
| Melt | -72.4 | 30.2 | 0.93 | **-1.35** | 0.32 |
| Sublimation | -1.6 | 0.2 | -0.20 | **0.00** | 0.00 |

### 5.2 Surface energy balance variability with climate

Figure 8 shows mean melt season (15th May to 30th September) fluxes for each year from 1958 to 2014. Shortwave and longwave radiation balances are relative small, 28 and -21 $\mathrm{W\,m}^{-2}$, respectively. Due to the late melt season onset over larger part of Svalbard little melt is occurring in May and first half of June. This results in a negative radiation balance even on sunny days due to high albedo and longwave cooling. $S_{net}$ and $Q_H$ are the main energy sources for melt, although all fluxes but $Q_G$ contribute during strong melt events, such as during summer cyclones. Over the period, there is a decrease in solar insolation ($S_\downarrow$) and an increase of $L_\downarrow$ indicating cloud thickening (Tab. 5). Despite reduced insolation, $S_{net}$ increases over the





**Figure 6.** Simulated specific climatic mass balance ($B_{\mathrm{clim}}$) averaged for the period 1957-2014 in $\mathrm{cm\,w.e.\,yr^{-1}}$

time period due to a decrease in albedo. Because of increased summer temperatures, $Q_H$ increases over the period, with a maximum during the warm 2013 melt season. Average $Q_L$ is close to zero in the 1960s, while it has been mostly positive since 1970. Both the turbulent fluxes have a significant increase over the period. During the 1960s mean $Q_M$ (13 $\mathrm{W\,m^{-2}}$) was less





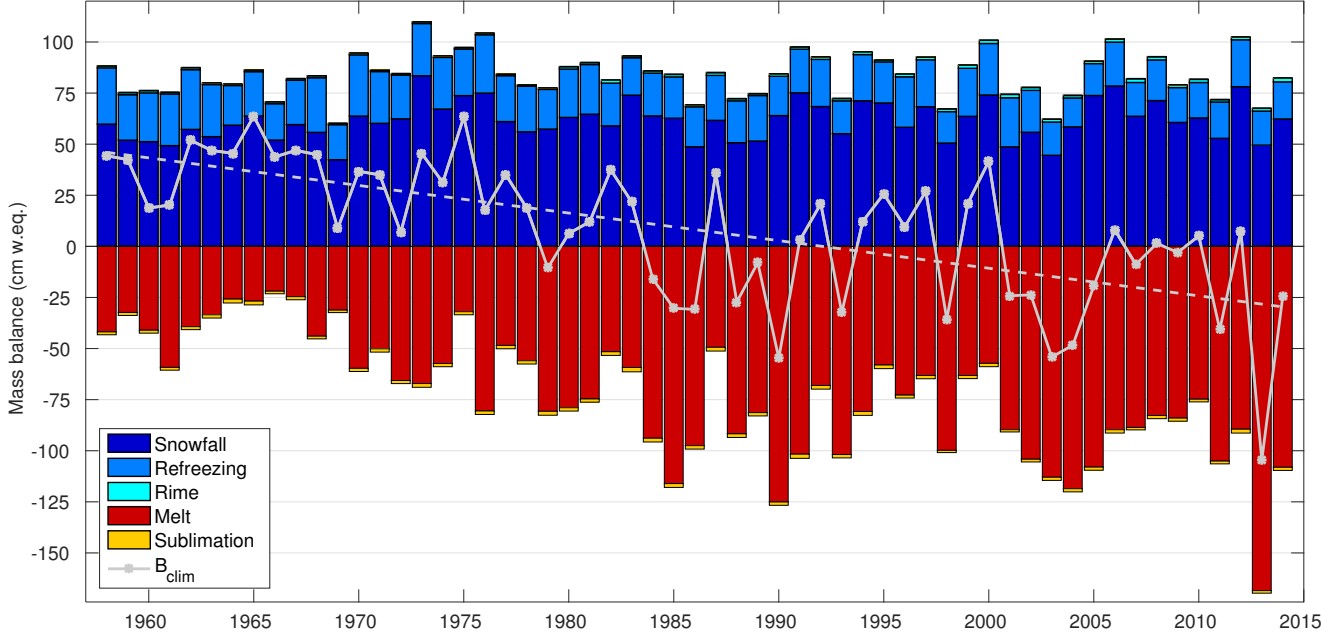

**Figure 7.** Yearly climatic mass balance ($B_{\mathrm{clim}}$) and its components, 1957-2014. Area averaging is performed using the temporal glacier mask. Dashed line is the linear mass balance trend of -14±4.1 cm w.e. decade$^{-1}$, with uncertainty referring to two standard deviations.

than half of the average after 2000 (30 W m$^{-2}$). $Q_G$ decreases over the period, which we associate with the reduction of snow and firn volumes. Glacier ice has a higher thermal conductivity than snow and firn, such that heat exchange is more efficient between the surface and the subsurface layers. In addition, higher conductivity during winter enables efficient cooling of the near-surface layers.

Annual sums of Positive Degree days ($\Sigma$PDD) increases from 82 during the 1960s to 213 Kd as 2000-14 average. Over the same period, the increase in melt is of similar magnitude, rising from 48 to 110 cm w.e. yr$^{-1}$. There is a very weak correlation ($R$=0.13) between winter snow accumulation and $B_{\mathrm{clim}}$. A much higher correlation of 0.55 is found between summer snowfall and $B_{\mathrm{clim}}$, which is linked to its effect on both albedo and refreezing. The correlation between PDD and snow accumulation is smaller (-0.40) such that the low temperatures alone cannot explain the summer snowfall effect. Of all

the glacio-meteorological variables in Tables 5 and 6, PDD and albedo are the quantities that can explain $B_{\mathrm{clim}}$ best. $Q_L$ is relatively small, but it is surprisingly well correlated to both melt and $B_{\mathrm{clim}}$. In the $Q_L$ calculation, humidity and temperatures in the air and at the surface are included together with wind speed, thereby integrating several relevant meteorological variables.

### 5.3  Refreezing and subsurface properties

Overall refreezing compromises 26 % of total accumulation, but the amount of refreezing is reduced by 1.2 cm yr$^{-1}$ decade$^{-1}$

over the period. With the slight increase of snowfall, the role of refreezing is reduced from 29 % of the accumulation in the 1960s to 22 % in the 2000s. The refreezing also has a profound effect on the subsurface thermal regime. When meltwater



**Figure 8.** Modeled energy fluxes ( $\mathrm{W\,m^{-2}}$ ) averaged over all glaciers and melt seasons (15th May- 30th Sept.) for the period 1957-2014. Area averaging was performed based on each year's glacier mask.

percolates and refreezes it releases large amount of latent heat; and at the same time, densification leads to an increase in thermal conductivity, thereby intensifying heat conduction.

Through refreezing and prolonged negative $B_{\mathrm{clim}}$, subsurface density and temperature change markedly over the period. Figure 9 shows that refreezing increases with altitude, with large interannual variability. During the cold 1960s there is decreasing refreezing with altitude above 500 m elevation. In this case, refreezing is limited by available water; Figure 9b indicates the presence of cold firn for these areas. After 1975 Svalbard accumulation areas are mostly temperate or near temperate at 15 m depth. Transition from cold to temperate firn is accompanied by a seasonal shift of the main refreezing period. In cold firn, most of the refreezing occurs when percolating water enters cold subsurface layers. While in the temperate firn, a large portion of the refreezing occurs when capillary water refreezes as the firn cools during winter. As the ELA increases, firn area extent




**Table 5.** Summary of modeled energy fluxes ( $\mathrm{W\,m^{-2}}$ ) averaged over all glaciers and melt seasons (15th May- 30th Sept.) for the period 1957-2014. STD is one standard deviation of the temporal variability, $R_{Q_M}$ is the correlation with $Q_M$ over the melt season, $\beta$ is the estimate of a linear trend over 1957-2014, and $\sigma_{slope}$ is slope uncertainty given by two standard deviation. Slope significance at the 95 %-level is in bold font ( $|\beta| > 2\sigma_{slope}$ ).

| Variable | Mean ( $\mathrm{W\,m^{-2}}$ ) | STD ( $\mathrm{W\,m^{-2}}$ ) | $R_{Q_M}$ (–) | $\beta$ ( $\mathrm{W\,m^{-2}\,yr^{-1}}$ ) | $2\sigma_{slope}$ ( $\mathrm{W\,m^{-2}\,yr^{-1}}$ ) |
|---|---|---|---|---|---|
| $S_\downarrow$ | 161 | 9.4 | 0.21 | -0.07 | 0.15 |
| $S_\uparrow$ | -133 | 9.2 | -0.62 | **0.24** | 0.13 |
| $L_\downarrow$ | 281 | 5.8 | -0.56 | 0.08 | 0.09 |
| $L_\uparrow$ | -303 | 3.0 | 0.60 | **-0.06** | 0.05 |
| $Q_H$ | 12.2 | 3.1 | -0.87 | **0.13** | 0.04 |
| $Q_L$ | 1.2 | 1.7 | -0.93 | **0.07** | 0.02 |
| $Q_G$ | -3.9 | 1.0 | 0.41 | **-0.03** | 0.01 |
| $Q_R$ | 0.1 | 0.1 | -0.80 | 0.00 | 0.00 |
| $\alpha$ (%) | 82.7 | 2.6 | 0.97 | **-0.11** | 0.03 |
| $Q_M$ | -22.6 | 8.9 | 1.00 | **-0.38** | 0.10 |

**Table 6.** Annual and melt season (15th May- 30th Sept.) means of glacio-meteorological variables averaged over all glaciers in Svalbard. STD is one standard deviation of the temporal variability, $R_{B_{clim}}$ is correlation with yearly $B_{clim}$, $\beta$ is the estimate of a linear trend, while $\sigma_{slope}$ is slope uncertainty given by two standard deviations. Slope significance at the 95 %-level in bold font ( $|\beta| > 2\sigma_{slope}$ ).

| Variable | Unit | Mean | STD | $R_{B_{clim}}$ | $\beta$ | $2\sigma_{slope}$ |
|---|---|---|---|---|---|---|
| Snow$_W$ | mm w.e. | 468 | 78.1 | 0.13 | **1.70** | 1.16 |
| Snow$_S$ | mm w.e. | 146 | 35.0 | 0.55 | **-0.65** | 0.53 |
| Rain$_W$ | mm w.e. | 23.2 | 14.9 | 0.01 | 0.05 | 0.24 |
| $T2_{annual}$ | °C | -9.00 | 1.5 | -0.49 | **0.07** | 0.02 |
| $T2_{summer}$ | °C | 0.23 | 0.8 | -0.85 | **0.03** | 0.01 |
| $T2_{winter}$ | °C | -12.1 | 1.9 | -0.41 | **0.08** | 0.02 |
| $\sum$PDD | d °C | 152 | 59.6 | -0.89 | **2.77** | 0.60 |
| $\alpha$ | % | 82.7 | 2.6 | 0.96 | **-0.11** | 0.03 |

is reduced and accordingly less potential for heat release through refreezing. These areas experience a cooling, starting around the ELA which varies from about 100 m a.s.l. in the northeast to 450 m a.s.l. in southern Spitsbergen. This results in an increase of cold ice area (Figure 9), which is predicted for polythermal glaciers in a warming climate, and also shown in model experiments (Irvine-Fynn et al., 2011; Wilson and Flowers, 2013). Since thermal regime changes depends on the ELA, which vary across Svalbard, the thermal response is better seen when resolved regionally as in Figure S6 and S7).



In contrast to the firn area, refreezing in the ablation area results in the formation of superimposed ice. Below the ELA the newly formed superimposed ice ablates later during the same melt season. Figure 10 shows yearly area-averaged refreezing, superimposed ice (SI) and internal accumulation which is refreezing below the previous summer surface. Over the period there is a decrease in internal accumulation following the firn area decrease. The loss of internal accumulation is only partly

5    compensated by refreezing above the previous summer surface. Thus, total refreezing decreases over the period, whereas the amount of SI formation in the SI-zone is quite stable, except in very negative years where nearly all formed SI ablates later om in the melt season.

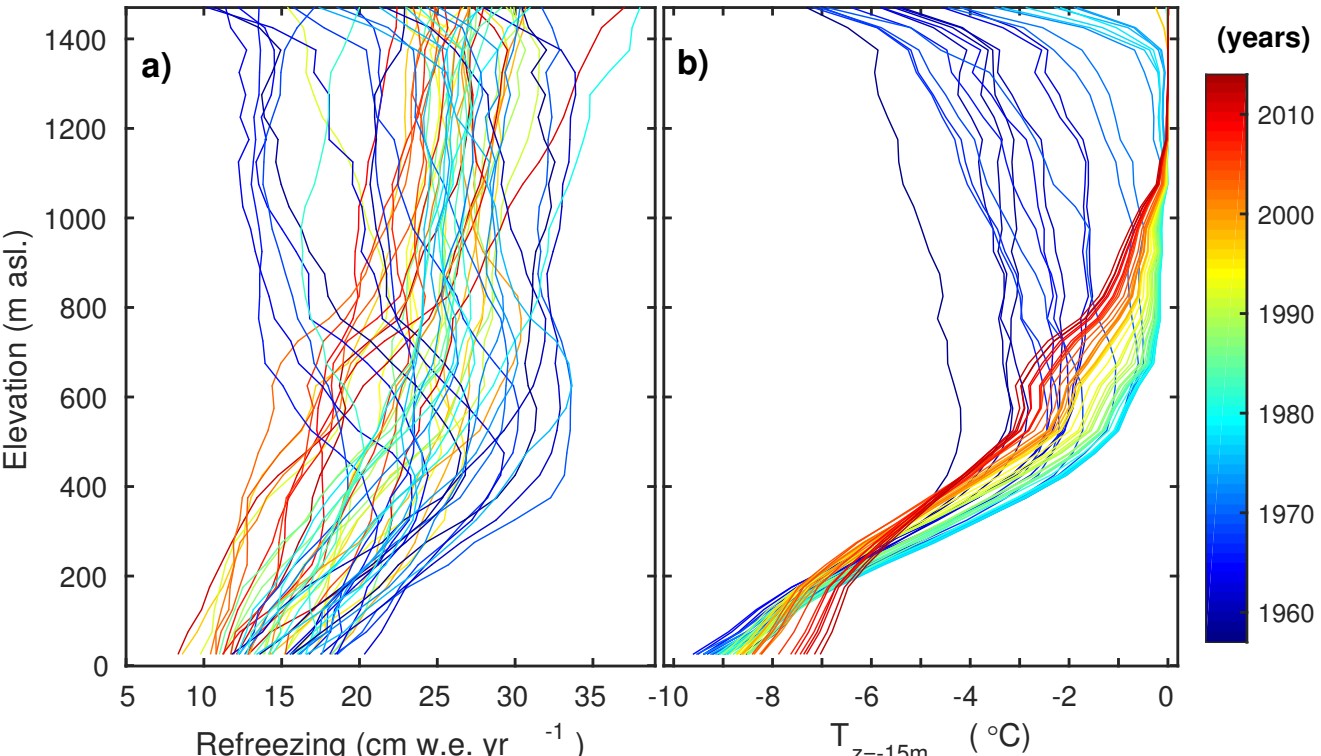

**Figure 9. a**) Refreezing rate and **b**) mean annual temperatures at 15 m depth averaged over 50 m altitude intervals from 1957 (blue) to 2014 (red).

The area covered by thick firn (dark blue, Fig. 10b) decreases over the period, except for small increases in the late 1960s and around 2000. Annual variability of glacier facies is almost exclusively due to the variability of the thin firn and superimposed

10   ice zone, with the thin firn area increasing during years of positive $B_{\mathrm{clim}}$ and vice versa. Similar fluctuations are seen for the firn bulk density, where the firnpack is densifying during years of negative $B_{\mathrm{clim}}$. Hence, the respective densities of the thin and thick firn reveal similar evolution as those of the firn area extent, with year-to-year fluctuations for the thin firn and damped decadal variations for the thick firn.



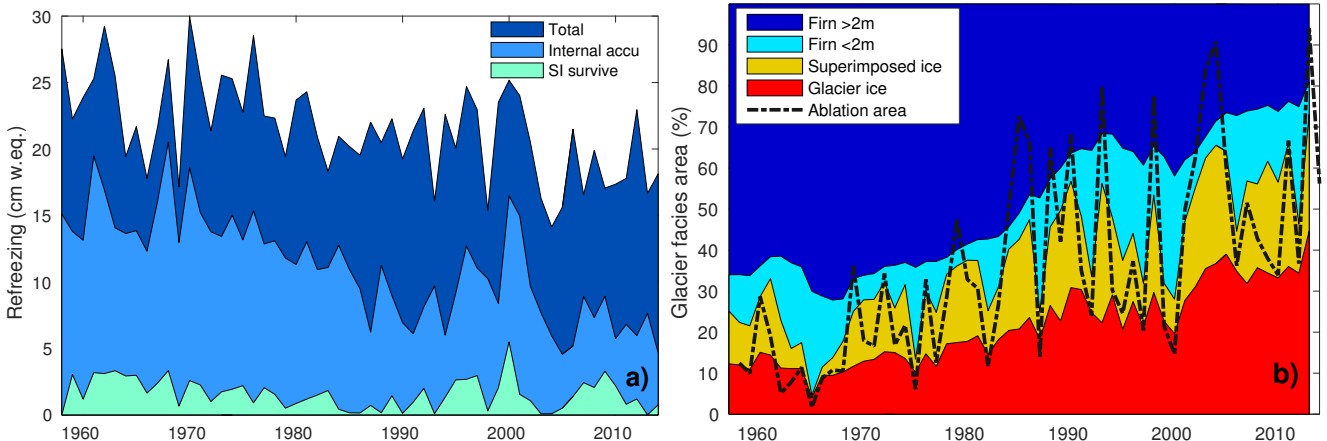

**Figure 10. a**) Annual total refreezing, superimposed ice end of melt season and internal accumulation (i.e. refreezing below the previous summer surface). **b**) Annual glacier facies divided into glacier ice, superimposed ice, thin firn (<2 m) and thick firn (>2 m).

## 5.4 Model validation and sensitivities

### 5.4.1 Mass balance from stakes and ice cores

The overall model performance in terms of $B_{\mathrm{clim}}$ is found by comparing stake mass balance readings and mass balance retrieved from ice cores with the model grid box closest to the measuring site. For the 1459 annual mass balances measurements, model

underestimates $B_{\mathrm{clim}}$ slightly by 1 cm w.e. yr$^{-1}$. However the much higher root mean square error (RMSE) of 59 cm w.e. yr$^{-1}$ reveals the existence of compensating errors. At Hansbreen, mass balance is underestimated by more than 100 cm w.e. yr$^{-1}$ in some years, while mass balance is overestimated at the lower stakes at the other three Spitsbergen glaciers. This is also illustrated in Figure 11, which also show that modeled mass balance gradients are too low for all five glaciers. RMSE is slightly improved (58 cm w.e. yr$^{-1}$) when excluding the 25 % of the mass balance measurements that are incorporated in the

calibration. The model performance is higher with this reduced dataset, since the importance of Hansbreen (the region with large underestimation) is reduced, as we assign equal weights to all measurements.

The model generally overestimates $B_{\mathrm{clim}}$ (Tab. 7) at the ice core sites, in contrast to the stakes. At the higher elevations of Holtedahlfonna, Nordenskiöldbreen, and Kongsvegen, Figure 11 shows that $B_{\mathrm{clim}}$ and winter balance are underestimated. Although mass balance stakes at these glaciers are in the proximity of the drilling sites, there is no overlap in time. Assuming

that both stake and ice core-derived mass balance measurements are correct, model error is not time invariant at these sites. Ice-core-derived mass balance at Holtedahlfonna increases over the period 1963-1991, while no change is found at Kongsvegen for this period (Tab. 7). Over the same period, modeled $B_{\mathrm{clim}}$ decreases slightly due to higher melt rates.

To test if there is a trend in the model performance we compare mass balance from Midtre Lovenbreen a small glacier southeast of Ny-Ålesund. We compare modeled $B_{\mathrm{clim}}$ from a pixel at 380 m elevation with measured mass balance at a

stake with corresponding altitude. There is good correlation between the two records, although the model overestimates $B_{\mathrm{clim}}$



**Figure 11.** Seasonal modeled and measured mass balance gradients (left panels) at the five validation glaciers. Dots and stars mark the position altitude of the stake for the measurements and corresponding DEM grid cell, respectively. Yearly ELA for the control run, measured and climate sensitivity test (right panels).

throughout the record. However, since 2000 there is hardly any bias, while $B_{\mathrm{clim}}$ is overestimated by about 40 cm w.e. yr$^{-1}$ for the period 1968-1979. This trend in model performance is driven by melt, and therefore most likely by summer air temperatures. Although $B_{\mathrm{clim}}$ is overestimated, winter accumulation is underestimated in the model which can be attributed to overestimation of sea ice in the reanalysis product at the northwest coast of Spitsbergen. The west coast is usually ice-free year round, but

5    prior to 1979 there were no satellite data to constrain sea ice cover and sea surface temperature in the reanalysis. An erroneous sea ice cover may compromise model heat and moisture uptake.





**Table 7.** Ice cores with derived climatic mass balance ($B_{\mathrm{clim}}$) ( cm w.e. yr$^{-1}$) from Pinglot et al. (1999, 2001), and error $\epsilon$ (modeled - measured) for the three time periods . Positive error if modeled is larger than measured.

| Location | Longitude | Latitude | Z | $Z_\epsilon$ | 1963-1986 | | 1986-199X | | 1963-199X | |
|---|---|---|---|---|---|---|---|---|---|---|
| | | | | | $B_{\mathrm{clim}}$ | $\epsilon$ | $B_{\mathrm{clim}}$ | $\epsilon$ | $B_{\mathrm{clim}}$ | $\epsilon$ |
| Kongsvegen-K | 13.3 | 78.8 | 639 | -39 | 50 | 6 | 48 | -8 | 48 | 7 |
| Kongsvegen-L | 13.4 | 78.8 | 726 | 17 | 59 | 14 | 62 | -11 | 60 | 10 |
| Snofjella-M | 13.3 | 79.1 | 1170 | -24 | – | – | 57 | 14 | – | – |
| Snofjella-W | 13.3 | 79.1 | 1190 | -44 | 37 | 52 | – | – | 47 | 39 |
| Vestfonna | 21.0 | 80.0 | 600 | 2 | 46 | 17 | 41 | 0 | 38 | 20 |
| Aust-98 | 24.0 | 79.8 | 740 | 12 | 48 | -4 | 52 | -24 | 50 | -10 |
| Lomonosov-76 | 17.5 | 78.8 | 1000 | 32 | – | – | – | – | 82 | 30 |
| Lomonosov-s8 | 17.5 | 78.8 | 1173 | -54 | – | – | 75 | 16 | – | – |
| Lomonosov-s10 | 17.4 | 78.9 | 1230 | 12 | – | – | – | – | 36 | 59 |
| Aasgaardsfonna | 16.7 | 79.5 | 1140 | -98 | – | – | – | – | 31 | 40 |
| Aust-F | 23.5 | 79.9 | 727 | -4 | – | – | 37 | 10 | – | – |
| Aust-D | 23.5 | 79.6 | 708 | -54 | – | – | 34 | 1 | – | – |
| Average | | | | | 48 | 17 | 48 | 1 | 49 | 24 |

## 5.5 Sensitivities

### 5.5.1 Parameters

Due to computational expense, sensitivity experiments are limited to stake locations on the five target glaciers over the period 2004-2013. In contrast to the calibration procedure, we here perturb one parameter at a time to isolate its effect. Table 8 show

5   parameters used in the control run, the perturbation and the $B_{\mathrm{clim}}$ response averaged over all stakes and years, relative to the control run ($dB_{\mathrm{clim}}$) and the geographical variability at all locations (SMB-stakes) in terms of standard deviation (STD). The sensitivity of ice albedo is surprisingly low in comparison to other parameters, while albedo aging parameters ($t*$) have rather large impact on $B_{\mathrm{clim}}$. This can be explained by the relative short exposure of glacier ice at the surface. Even in the ablation area, snowfall is common during the melt season, such that the rate of albedo decay is more important than the actual threshold

10   values. Modeled $B_{\mathrm{clim}}$ is robust with respect to the choice of roughness lengths ($zo$). A 1 K change in the rain-snow threshold temperature has a comparable effect as a 10 % precipitation change. More than 90 % of this change is taking place during the melt season. Although a large portion of the precipitation occurs at mild temperatures, the effect of the rain-snow threshold on winter mass balance is not so important since most of the winter rain refreezes.





**Table 8.** Parameter sensitivity demonstrated by perturbations impact on $B_{\text{clim}}$ where $dB_{\text{clim}}$ (cm w.e.) is the difference from the control run averaged over all stakes and years, STD ( cm w.e.) is the variability of the difference for both year-to-year and site-to-site.

| Parameter | control | perturbation | Response | |
|---|---|---|---|---|
| | | | $dB_{\text{clim}}$ | STD |
| $\alpha_{ice}$ | 0.3 | +0.05 | 0.4 | 6.1 |
| | | -0.05 | -0.7 | 6.9 |
| $\alpha_{snow}$ | 0.85 | + 0.05 | 5.6 | 6.7 |
| | | - 0.05 | -6.4 | 7.5 |
| $\alpha_{firn}$ | 0.62 | + 0.05 | 7.1 | 7.2 |
| | | - 0.05 | -7.1 | 7.5 |
| $t*$ | $\{$ 5,15,100 d $\}^{a}$ | ×0.5 | -8.3 | 7.7 |
| | | ×2 | 9.4 | 7.3 |
| $zo$ | $\{0.18,0.06$ mm $\}^{b}$ | ×0.5 | 6.5 | 8.2 |
| | | ×0.25 | 12.7 | 11.2 |
| | | ×2 | -7.1 | 7.8 |
| | | ×4 | -14.0 | 11.3 |
| $T_{rain/snow}$ | 1.5 | -1K | -3.8 | 8.1 |
| | | +1K | 7.9 | 9.3 |
| Temperature | | +1K | -29.7 | 20.8 |
| | | +2K | -64.7 | 38.7 |
| | | +3K | -109.0 | 61.3 |
| | | -1K | 29.9 | 18.7 |
| | | -2K | 61.3 | 35.0 |
| | | -3K | 87.7 | 52.1 |
| Precipitation | | +15% | 13.1 | 8.0 |
| | | +30% | 25.2 | 9.4 |
| | | -15% | -14.2 | 7.6 |
| | | -30% | -29.7 | 10.4 |
| Førland (2011)$^{c}$ | | +4K & +5% | -81.7 | 49.4 |
| Førland (2011)$^{d}$ | | +6K & +30% | -134.4 | 87.4 |

[a] time scales of aging is 5,15 and 100 days at temperatures of 0 °C (wet),0 °C (dry) and -10 °C.

[b] roughness lengths for ice and snow respectively.

[c], [d] Climate scenario for western and northeastern Svalbard respectively, as in Førland et al. (2011)





### 5.5.2 Climate

Climate sensitivity is performed in the same manner as for parameters, but now by perturbing temperature and precipitation, without seasonality by, $\pm 1$, $\pm 2$ and $\pm 3$ K for temperature and by $\pm 15$ % and $\pm 30$ % for precipitation. A temperature increase of 1 K result in a $B_{\mathrm{clim}}$ decrease of 30 cm w.e., same as found by van Pelt et al. (2012). The temperature sensitivity of roughly

-30 cm w.e. yr$^{-1}$ K$^{-1}$ is in the lower range reported from glaciers elsewhere in the world (see De Woul and Hock (2005) and references therein). Although a comparably small temperature sensitivity, the low mass flux turn over in Svalbard makes this sensitivity equal or even higher than other glacierized regions in terms of its impact on $B_{\mathrm{clim}}$. This is exemplified by comparing the impact of a temperature perturbation to the mass balance components of the control run over the 2004-13 period, where a temperature increase of 1 K results in a 30 % increase in ablation, and a more than doubled mass loss. A precipitation increase

of nearly 40 % is necessary to compensate a temperature increase of 1 K.

We also perform climate experiments with temperature and precipitation predictions for 2100 (Førland et al., 2011). Since changes in temperature and precipitation at Svalbard are projected to occur with a substantial southwest-northeast gradient, we perform two experiments. In the first scenario, applicable for western Svalbard, we increase annual temperature and precipitation by 4 K and 5 %, respectively. In the second scenario, applicable For northeastern Svalbard we use an annual increase of

6 K and 30 %, respectively. For these future scenarios we apply seasonality in air temperature, as suggested in Førland et al. (2011), with summer temperature increase of 2.5 K for western Svalbard and 4 K for the northeastern Svalbard. Figure 11 shows that ELA on the target glaciers is mostly above present topography under both scenarios. Note that the predictions of Førland et al. (2011) are for 2071-2100 averages relative to the normal period 1961-1990. We apply these increases on the 2004-2013 period which have an annual air temperature of -3.2 °C, already 3.5 K warmer than the normal period (1961-1990).

### 5.5.3 Glacier mask

We test the impact of glacier masks on $B_{\mathrm{clim}}$ by applying the three different fractional glacier masks (Sec. S2); the reference mask (all-time max), the 2000s mask and the time-varying mask. $B_{\mathrm{clim}}$ for entire Svalbard is 4.5 cm w.e. yr$^{-1}$ for the reference mask, 10 cm w.e. yr$^{-1}$ for the 2000s mask and 8.2 cm w.e. yr$^{-1}$ for the time-varying mask. Despite relatively small differences in area and specific mass balance, there is a 100 Gt ice mass difference over the 56 year period for the different glacier masks.

In comparison this is more than a third of the annual contribution of all glaciers and ice caps to sea level rise. Since the area difference between the glacier mask mainly occurs in the lower ablation zone, where $B_{\mathrm{clim}}$ fluxes are largest, a small area change has a substantial impact on the glacier wide $B_{\mathrm{clim}}$. Hence, representations of glacier margins in $B_{\mathrm{clim}}$ models are very important. Mass balance using the time-varying mask represents something in between the conventional and reference surface mass balance (Elsberg et al., 2001), since glacier area is annually updated while the DEM is static.



## 6 Discussion

### 6.1 Controls on $B_{\mathrm{clim}}$ at present and in a future warmer climate.

Variability in annual $B_{\mathrm{clim}}$ is found to be dominated by summer melt, thereby confirming other studies at Svalbard (Lang et al., 2015a; van Pelt et al., 2012). Melt variability is driven by melt season air temperatures. Albedo, which in Table 4 and 9 shows the largest correlation with $B_{\mathrm{clim}}$, is closely related to net short wave radiation, the largest energy source for melt. Albedo is also indicative for winter snow and summer snowfall events since snow is brighter than bare ice (Tab. 9), and temperature which controls both the precipitation phase as well as the rate of albedo decay.

With a future climate as projected by Førland et al. (2011) increased precipitation can only partly compensate $B_{\mathrm{clim}}$ for enhanced melting. Lang et al. (2015b) argue that changes in future cloud cover will reduce the negative effect of a warming climate on $B_{\mathrm{clim}}$. Statistics from our simulations indicate that the net radiation is positively correlated with summer air temperatures (Tab. 9), corresponding to a dominance of longwave radiation over shortwave radiation. However, cloud cover is rather uncertain in our data in combination with extrapolation to the future makes this interpretation rather speculative.

As on Greenland, melt water retention through refreezing on Svalbard have been proposed as an efficient buffer for mass losses in a future warmer climate (Harper et al., 2012; Wright et al., 2005). However, here we find that refreezing and retention decrease over 1957-2014 period, both in absolute numbers and as a percentage of the $B_{\mathrm{clim}}$. These findings are in line with other studies on Svalbard (Van Pelt and Kohler, 2015) and Greenland (Charalampidis et al., 2015; Machguth et al., 2016). Even if the superimposed ice area increases, at the expense of the firn area, superimposed ice formation does not compensate for the loss of internal accumulation. In contrast to Greenland, the existing area of cold firn in Svalbard is not large enough to buffer effects of future warming. Furthermore, our model is in its current formulation not capable of reproducing impermeable ice layers within the firn; such layers would further decrease the available storage volume by diverting runoff laterally (Mikkelsen et al., 2015).

Despite an overall decrease in refreezing, some low altitude areas along the western coast show an increase due to winter rain events (e.g. Hansen et al., 2014). During and in the first days after rain events substantial amount of superimposed ice form. However, for the remaining winter, densification of the snowpack reduces the the insulating effect such that the initial latent heat release is largely depleted through intensified cooling.

For a warmer climate, our results suggest a cooling of glacier ice close to the (rising) ELA. At the glacier front, modeled temperatures at 15 m depth increased by 2-4 °C over 1957-2014. It is likely that glaciers which usually have their snouts frozen to the ground will approach melting point in near future if warming continues. Frozen glacier fronts may act as a plug for the upstream glacier flow, and provide a mechanism to slow the entire glacier (e.g. Dunse et al., 2015). Furthermore, a thermal switch at the glacier base allows sliding and is proposed as surge initiation mechanism (e.g. Murray et al., 2000). Historical records shows that surge frequency at Svalbard is connected with periods of warming and negative $B_{\mathrm{clim}}$ (W. Farnsworth, 2015, Pers. comm., October 31). Increased ice flow leads to more crevassing, which again promote cryo-hydrological-warming and basal lubrication, acting as a positive feedback in the system (Phillips et al., 2010; Dunse et al., 2015). Although these mechanism are not fully understood, theory suggests that increased runoff and changes in the thermal and hydrological regimes





may trigger widespread changes in the velocity structure of large ice masses. To further study the coupling between surface and basal processes, melt rates and near-surface temperatures must be reliably quantified; this requirement emphasizes potential further use of the dataset resulting from our study.

## 6.2 Comparison to other studies

To facilitate comparison, we extract subsets of our results that best correspond to mass balance estimates by other studies in terms of time period and area (see Tab. 10). Nevertheless, they are not all directly comparable since we employ $B_{\mathrm{clim}}$ for hydrological years while the geodetic studies may start one season and end in another, different area coverage where some studies do not state whether Kvitøya is included, or if some regions are largely unrepresented. Furthermore, different methods measure different parts of the total glacier mass balance. For example, we estimate $B_{\mathrm{clim}}$ only partly accounting for glacier area

changes, since the mass loss of the changing margins is not included. Geodetic approaches (Moholdt et al., 2010; Nuth et al., 2010) include calving from dynamics, but not from frontal position changes. Gravimetrical estimates by GRACE includes all components of the glacier mass balance.

Figure 12 shows mass balance through time compared with studies summarized in IPCC5 along with three other recent $B_{\mathrm{clim}}$-studies. Average mass balance for the respective studies are also listed in Table 10. Overall there is a good agreement

with the other studies, but our $B_{\mathrm{clim}}$ estimates are more negative than the others for the latter part of the study period whereas the opposite is true during the first part. Our estimate must be higher than the geodetic and gravimetric estimates since they account for mass loss by calving, which is estimated to be $13\pm5\,\mathrm{cm\,w.e.\,yr^{-1}}$ Błaszczyk et al. (2009). Compared to Nuth et al. (2010), our estimate is $30\,\mathrm{cm\,w.e.\,yr^{-1}}$ less negative which seems to be caused by too positive $B_{\mathrm{clim}}$ in northeastern Spitsbergen. We attribute the $B_{\mathrm{clim}}$ overestimation in these areas to overestimation of precipitation and too low air temperatures in our forcing

dataset. The north-south gradient in downscaled mean annual air temperatures is twice as large as that observed at weather stations in Hornsund, Longyearbyen and Ny-Ålesund. The exaggerated horizontal temperature gradient (Figure 4b) is possibly linked to incorrect position of the sea ice edge in the ERA reanalysis (Aas et al., 2016). During summer, observations show that the southernmost station (Hornsund) is coldest, since the other two have a more continental climate. Continental warming in broad ice-free valleys is not well captured by the downscaling, due to a mismatch of the sea-land-glacier mask between the

coarse reanalysis grid and the finer downscaled grid. This results in too low temperatures at glacier fronts in the interior of Spitsbergen and consequent underestimation of ablation rates.

## 6.3 Uncertainties

Uncertainties are introduced at every step in the process chain and accumulate in the simulated $B_{\mathrm{clim}}$. Uncertainties comprise the climate forcing and initial state, model set-up and parameterizations, topographic simplification, and uncertainties of the

measurements used for calibration. However, it is hard to quantify the contribution of the respective sources, especially since the calibration may compensate for systematic errors.

Glaciological surface mass balance represents a useful quantity for validating our model. Not only because it resembles our goal, mass balance, but it is sensitivite to atmospheric and surface conditions over a large variate of temporal and spatial scales,





**Table 9.** Correlation coefficients between annual mass balance components, average melt season surface energy fluxes, and selected meteorological variables. Precipitation is separated into snow and rain where indices $s$ refers to summer averages (15 May - 31 Sep.) and $w$ refers to the period 1 Oct-14 May. $T_A$ and $T_S$ are annual and summer averages of near-surface air temperature over the entire glacier area, respectively.

| | $B_\mathrm{clim}$ | $B_\mathrm{ACCU}$ | $B_\mathrm{ABLA}$ | $B_\mathrm{ref}$ | $Q_N$ | $S_\downarrow$ | $L_\downarrow$ | $Q_H$ | $Q_E$ | $Q_G$ | $\alpha$ | Snow$_w$ | Snow$_s$ | Rain$_w$ | $T_A$ | $T_S$ | $\Sigma PDD$ |
|---|---|---|---|---|---|---|---|---|---|---|---|---|---|---|---|---|---|
| $B_\mathrm{clim}$ | 1.00 | 0.48 | 0.95 | 0.62 | -0.31 | 0.05 | -0.46 | -0.78 | -0.86 | 0.35 | 0.96 | 0.13 | 0.55 | 0.01 | -0.49 | -0.85 | -0.89 |
| $B_\mathrm{ACCU}$ | - | 1.00 | 0.19 | 0.53 | -0.61 | -0.39 | -0.10 | 0.02 | -0.15 | -0.13 | 0.36 | 0.84 | 0.60 | 0.27 | 0.27 | -0.20 | -0.12 |
| $B_\mathrm{ABLA}$ | - | - | 1.00 | 0.51 | -0.14 | 0.19 | -0.48 | -0.88 | -0.91 | 0.43 | 0.95 | -0.15 | 0.41 | -0.09 | -0.64 | -0.88 | -0.95 |
| $B_\mathrm{ref}$ | - | - | - | 1.00 | -0.35 | -0.06 | -0.33 | -0.37 | -0.46 | -0.05 | 0.59 | 0.10 | 0.37 | 0.30 | -0.30 | -0.53 | -0.50 |
| $Q_N$ | - | - | - | - | 1.00 | 0.79 | -0.06 | -0.01 | 0.13 | 0.02 | -0.31 | -0.38 | -0.65 | -0.09 | -0.13 | 0.07 | 0.07 |
| $S_\downarrow$ | - | - | - | - | - | 1.00 | -0.66 | -0.16 | -0.17 | -0.05 | 0.10 | -0.36 | -0.35 | -0.13 | -0.36 | -0.37 | -0.30 |
| $L_\downarrow$ | - | - | - | - | - | - | 1.00 | 0.24 | 0.43 | 0.11 | -0.54 | 0.12 | -0.22 | 0.10 | 0.43 | 0.68 | 0.58 |
| $Q_H$ | - | - | - | - | - | - | - | 1.00 | 0.92 | -0.52 | -0.79 | 0.27 | -0.22 | 0.01 | 0.59 | 0.68 | 0.87 |
| $Q_E$ | - | - | - | - | - | - | - | - | 1.00 | -0.40 | -0.87 | 0.17 | -0.40 | -0.01 | 0.54 | 0.76 | 0.89 |
| $Q_G$ | - | - | - | - | - | - | - | - | - | 1.00 | 0.43 | -0.24 | 0.27 | 0.03 | -0.41 | -0.35 | -0.45 |
| $\alpha$ | - | - | - | - | - | - | - | - | - | - | 1.00 | -0.04 | 0.59 | -0.02 | -0.58 | -0.89 | -0.94 |
| Snow$_w$ | - | - | - | - | - | - | - | - | - | - | - | 1.00 | 0.20 | 0.25 | 0.59 | 0.15 | 0.23 |
| Snow$_s$ | - | - | - | - | - | - | - | - | - | - | - | - | 1.00 | -0.06 | -0.25 | -0.43 | -0.41 |
| Rain$_w$ | - | - | - | - | - | - | - | - | - | - | - | - | - | 1.00 | 0.44 | 0.07 | 0.15 |
| $T_A$ | - | - | - | - | - | - | - | - | - | - | - | - | - | - | 1.00 | 0.63 | 0.73 |
| $T_S$ | - | - | - | - | - | - | - | - | - | - | - | - | - | - | - | 1.00 | 0.91 |
| $\Sigma PDD$ | - | - | - | - | - | - | - | - | - | - | - | - | - | - | - | - | 1.00 |




**Table 10.** Svalbard mass balance estimates from different studies and methods. Note that the numbers between the studies are not directly comparable due to different time periods, area cover and methodology, see text for explanation.

| Period | this study | Other studies | | |
|---|---|---|---|---|
| | cm w.e. yr$^{-1}$ | cm w.e. yr$^{-1}$ | Method | Reference |
| 2003-2008 | -13.4 | -12±4 | ICESat | Moholdt et al. (2010) |
| 2003-2010 | -9.3 | -9±6 | GRACE | Jacob et al. (2012) |
| 2003-2008 | -20.2 | -34±19 | GRACE | Mèmin et al. (2011) |
| 2003-2009 | -11.6 | -13±6 | Mix | Gardner et al. (2013) |
| 2003-2013 | -20.6 | -25.7 | WRF-$B_{clim}$ | Aas et al. (2016) |
| 2000-2011 | -18.8 | -5±40 | RCM-$B_{clim}$ | Möller et al. (2016) |
| 1979-2013 | -7.8 | -5.4 | MAR-$B_{clim}$ | Lang et al. (2015a) |
| 1965/71/90-2005[a] | -5.8 | -36±20 | Geodetic | Nuth et al. (2010) |
| 1970-2000 | 8.5 | -1.4±0.3 | Mix (SMB) | Hagen et al. (2003b) |
| 1970-2000 | 8.5 | -27±33 | Mix (SMB) | Hagen et al. (2003a) |

[a] start of time series depends on location, see Nuth et al. (2010) for details.

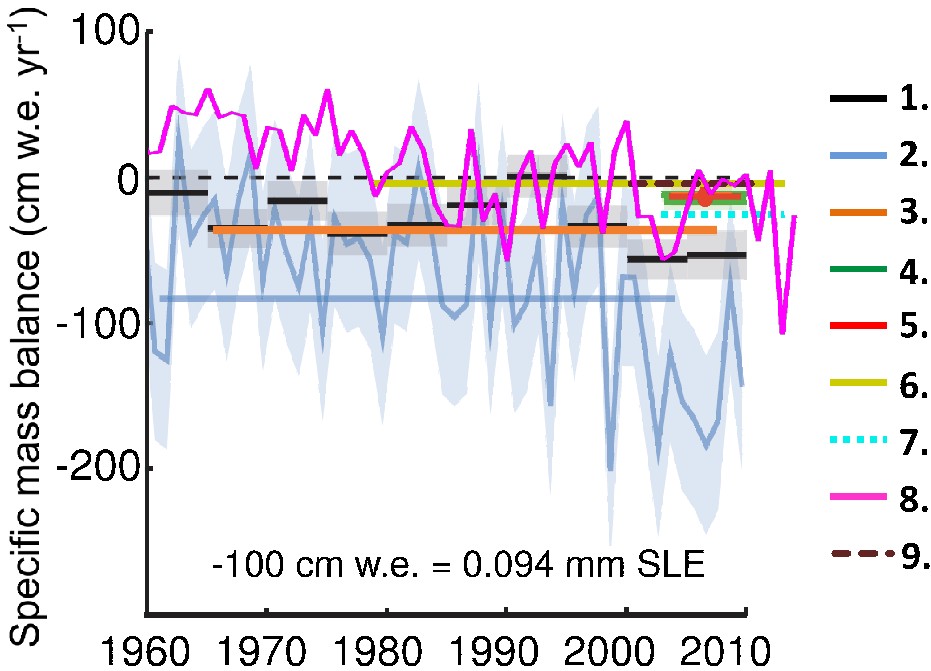

**Figure 12.** Glacier mass balance estimates for Svalbard, 1) interpolation of local records, 2) modelling Marzeion et al. (2012), 3) gravimetry, 4) geodetic, 5) mix Gardner et al. (2013), 6) modelling Lang et al. (2015a), 7) modelling Aas et al. (2016), 8) modelling Möller et al. (2016), 9) this study. Shaded area of 1) and 2) indicate the uncertainty. Figure modified from Vaughan et al. (Fig. 4.11. 2013)





thus ideal for validating land surface models in general. Since subsurface properties (temperature, density and water content) are usually not well captured by glaciological mass balance measurements, the measurement uncertainty is higher in the firn areas compared to the $10\,\mathrm{cm\,w.e.\,yr^{-1}}$ uncertainty for glacier ice areas (Sec.4.4).

Model performance at the validation sites (ice cores and stakes) is satisfactory with a mean bias of $-1\,\mathrm{cm\,w.e.\,yr^{-1}}$, but
substantial compensating errors exist, as indicated by the RMSE of $59\,\mathrm{cm\,w.e.\,yr^{-1}}$. For Svalbard as a whole, $B_{\mathrm{clim}}$ is in agreement with other studies, but regionally $B_{\mathrm{clim}}$ is overestimated in northern Spitsbergen and underestimated in southern Spitsbergen.

Uncertainties in the climate forcing are probably the main source of error. We substantiate this statement by the importance of air temperature on $B_{\mathrm{clim}}$ ($-30\,\mathrm{cm\,w.e.\,yr^{-1}\,K^{-1}}$), the low sensitivity of model parameters (Sec. 5.5), and that the model
is capable of reproducing measured $B_{\mathrm{clim}}$ when forced with local climate data at the weather station site (Østby et al., 2013). Validation at weather stations indicates that our climate forcing is of similar or slightly lower quality as other, more expensive downscaling studies (Claremar et al., 2012; Lang et al., 2015a; Aas et al., 2016).

Downscaled summer air temperatures are in general too low at the coastal weather stations (Tab. 2). In contrast, downscaled air temperatures are too warm at the glacier weather stations. Theses biases, combined with the underestimation of summer
mass balance gradients (Fig. 11), indicate too low air temperature lapse rates in the downscaling. Since satellite-derived surface temperatures are capped at the melting point, only temperature measurements at elevation substantially higher than the ERA-orography can resolve this issue. The underestimated summer mass balance gradient may, at least partly, be explained by underestimated winter mass balance gradients (Fig. 11), through its effect of prolonged snow cover and increased albedo and refreezing.

Downscaled precipitation compares well to measurements at the coastal stations (Sec. 3.2.1) and seasonal precipitation is mostly well reproduced over the glaciers when compared to ice cores and winter mass balance from stakes and ground penetrating radar. Underestimation of winter mass balance gradients (Fig. 11) are not necessarily indicative of to low orographic enhancement in the LT-model, but could be caused by wind redistribution. Snowdrift accumulate in concave-shaped accumulation areas, while the wind erode ablation areas that tend to have a convex-shaped surface topography. In contrast, ice cores
indicates that precipitation is overestimated at higher elevation in northern Spitsbergen. This bias could be caused by the wind exposed ice core drilling sites at ice field summits. The largest precipitation underestimation (up to $100\,\mathrm{cm\,w.e.\,yr^{-1}}$) is found at Hansbreen, which is known to have an asymmetrical snow accumulation across the centerline due to wind redistribution (Grabiec et al., 2006).

The TopoSCALE methodology for downscaling climate variables shows several shortcomings when applied to Svalbard.
In the Alps, where the methodology was developed, differences between the coarse reanalysis data and the finer grid for downscaling are governed by elevation difference between the coarse and fine grid. On Svalbard, large horizontal gradients of atmospheric heat and moisture arise from the interaction between open water, sea ice, tundra and glacier-covered areas, in addition to the vertical gradients. Due to the coarse spatial resolution of the reanalysis, some land areas of our finer topography may be wrongly represented as ocean in the reanalysis, thereby considerable affecting the downscaled variables. For instance,
land areas of south Spitsbergen are largely represented as ocean in the ERA land mask, such that sea surface temperatures are





incorporated into the downscaling of air temperatures over land and glaciers in the fine scale topography. This gives rise to considerable biases and thereby also affects temperature gradients in the downscaled temperature field. A similar effect arises if an erroneous sea ice mask is employed in the reanalysis. The climate reanalysis is a quite homogeneous product in time, except for sea surface temperatures and sea ice cover, which was substantially improved with the advent of satellite-borne sensors,

and further improved with newer sensors. We link these improvements over time with the higher climate forcing quality after 1980 (Sec. 3.2.1).

Calculations of turbulent fluxes and albedo are the processes within the model which have the highest sensitivities regarding parameter uncertainty on modeled $B_{\mathrm{clim}}$. Model performance is quite robust to the choice of roughness lengths used to calculate turbulent fluxes, but values typically span several orders of magnitude in the literature. Albedo parameterization is shown to be

more sensitive to the aging parameters rather than the actual threshold values used in the parameterization. The applied albedo parameterization does not account for spatial variability due to impurity content, which is possibly a major weakness given that ice albedo vary from 0.15-0.44 across Svalbard (Greuell et al., 2007). Dark bands such as in western Greenland (Wientjes et al., 2011) are also observed in Svalbard, but are not included in the model. Extending the model albedo formulation to account for dust and impurity content would be a subject for future work. Model parameters concerning runoff and water retention may

also be a significant error source, but we lack observational data to evaluate this aspect. The bucket-type water percolation method is also questionable, as infiltration is highly heterogeneous and horizontal fluxes are neglected (e.g. Reijmer et al., 2012; Cox et al., 2015). A 10 year spin-up time is found to be sufficient for the model performance, although sites close to the ELA showed a larger sensitive to the spin-up period in the first years of the run (<5 cm w.e.). From the Svalbard temperature record, we know that the 1960s were colder than the 1950s. Using the cold 1960s as a spin-up period has likely led to a too

large potential for refreezing given the lower temperatures of the early period. Additionally, colder temperatures cause less dense firn and further exaggerate the retention potential.

Although the 1000 m resolution DEM applied in this study largely reproduces the hypsometry of the 90 m DEM, glacier mask sensitivity reveal differences of 1.8 $\mathrm{Gt\,yr^{-1}}$ between different glacier masks (Sec.5.5.3), slightly lower than the 2.1 $\mathrm{Gt\,yr^{-1}}$ mass loss from tidewater margin retreat (Błaszczyk et al., 2009). We asses possible errors introduced by using a static

DEM by considering a typical mass balance gradient of 0.2-0.3 $\mathrm{cm\,w.e.\,m^{-1}}$. A uniform elevation change of 10 m for the entire glacier surface would alter the mass balance by 2 $\mathrm{cm\,w.e.\,yr^{-1}}$, equivalent to 1 $\mathrm{Gt\,yr^{-1}}$. Observed elevation changes in southern and western Spitsbergen for 1930s to 1990s are typically a few tens of meters (Nuth et al., 2007). In the early part of the study period, when DEM difference is largest, this error might amount up to 1 $\mathrm{Gt\,yr^{-1}}$.

# 7   Conclusions

This study presents modeled climatic mass balance for all glaciers in Svalbard for the period 1957-2014. Modeled mass balance is mostly in line with similar studies, where comparisons are possible. Despite overall good model performance, validation with in-situ mass balance measurements indicate regionally compensating errors. Our main findings are:



- Svalbard $B_{\mathrm{clim}}$ is estimated to 8.2 $\mathrm{cm\,w.e.\,yr^{-1}}$, corresponding to a mass surplus of 175 Gt over the 1957-2014 period. Mass loss increases over the period and $B_{\mathrm{clim}}$ switches from a positive to negative regime around 1980, with a trend of -1.4±0.4 $\mathrm{cm\,w.e.\,yr^{-2}}$. For the ERA-interim period 1979-2014, there is $B_{\mathrm{clim}}$ trend of -0.96±0.99 $\mathrm{cm\,w.e.\,yr^{-2}}$, which is not significant at the 95 %-level. Current $B_{\mathrm{clim}}$ for the period 2004-2013, combined with frontal ablation estimates (Błaszczyk et al., 2009), yields a total Svalbard mass balance of -39 $\mathrm{cm\,w.e.\,yr^{-1}}$, which corresponds to a eustatic sea level rise of 0.037 $\mathrm{mm\,yr^{-1}}$.

- There is large interannual variability in $B_{\mathrm{clim}}$, which is controlled by summer melt. Decreasing $B_{\mathrm{clim}}$ over the study period is primarily due to increased summer temperatures, amplified by the albedo feedback.

- Refreezing plays a major role in Svalbard $B_{\mathrm{clim}}$, representing about a quarter of the accumulation. As the climate gets warmer over the study period, refreezing and water retention decreases both in absolute and relative percentage of $B_{\mathrm{clim}}$.

- Following equilibrium line rises, firn extent is reduced and subsurface cooling occurs, while subsurface warming occurs lower on the glaciers. Increased runoff and changes in the hydrological and thermal regimes are likely to be important for glacier flow.

- Sensitivity experiments show that future increased precipitation can not compensate for increased air temperatures. Perturbing current climate with future scenarios from a RCM (Førland et al., 2011), results in modeled ELA above the summit of most of today's ice caps and ice fields.

Two major shortcomings in the meteorological forcing are identified. Firstly, climate prior to 1980 is too cold and dry, possibly caused by sea ice representation in the reanalysis data. Sea ice is poorly constrained in the period prior to 1979 when satellite data were not available. Secondly, the downscaling methodology likely introduces biases where the surface type (ocean, sea ice, tundra, glacier) does not match across the scale gap between the coarsely resolved reanalysis and our finer grid resolution. This problem may be overcome by using the same downscaling approach on 10-25 km RCM output instead of the ERA-40/ Interim data, which could potentially improve our results.

In addition to considerable mass change, this study suggests there are significant changes in the thermal and hydrological regimes of Svalbard glaciers, which in turn may have important implications for glacier dynamics as suggested by resent observation of ice cap destabilization Dunse et al. (2015); an understudied process that requires further investigation.

*Acknowledgements.* The Norwegian Ministry of Education funded a scholarship for T. Østby. Fieldwork and analysis have been funded by ESA (CRYOVEX), NFR TIGRIF, EU Ice2Sea, NFR CRYOMET and the NCoE SVALI. The authors gratefully acknowledge the enthusiastic support by T. Dunse, T Eiken, G. Moholdt and the Austfonna team during fieldwork. We further wish to thank C. Nuth for providing glacier outlines and Digital Elevation Models, J. Vrugt for sharing the DREAM-code and defining the likelihood function. Mass balance from Hornsund were provided by B. Luks and from Nordenskiöldbreen by V. Pohjola and W. van Pelt. We are also grateful for a number of meteorological data: ERA reanalysis provided by the European Centre for Medium-Range Weather Forecasts, weather station measurements provided by the Norwegian meteorological office through eklima.no, and by the University Centre of Svalbard through unis.no/resources/weather-stations-and-web-cameras/. Radiation from the BRSN-station in Ny-Ålesund are provided by AWI through Pangaea.de



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
