# Peer review of "Diagnosing the decline in climatic mass balance of glaciers in Svalbard over 1957-2014."

_The Cryosphere, 2016_

## Referee Comment (RC1) · X. Fettweis (Referee) · 29 Sep 2016

This paper presents estimation of SMB at 1km of resolution over Svalbard downscaling directly the ERA reanalysis to 1km as in Hanna et al. (2008) over GrIS. This papers is generally well written and deserves to be published in TC although the main conclusions of this paper (lines 7-16, pg 32) just confirm previous results (e.g. Lang et al, 2015) and do not really bring new stuff. However, the methodology used is different than previous studies and gives results at 1km which will be very useful afterwards to force ice sheet models. Nevertheless, before publication, some issues need to be resolved if it is not a too big job for the authors.

Major:

- line 2, pg 32 and abstract: the authors suggests a change of SMB around 1980 which

corresponds to the switch from ERA40 to ERAint. Discontinuities in SST/SIC are also mentioned around 1980 in the manuscript. How does this discontinuity impact on the results presented here? This change should be more discussed because it could be just an artefact of the use of inhomogeneous dataset. How do the results compare over 1980-2001 when both reanalysis are available? Estimation using ERA-40 over 1979-2001 should be added in this manuscript to check the homogeneity (assumed in the manuscript without proof: pg 31, line 3) of this SMB reconstruction over 1957-2014.

Minor:

- line 26, pg 4: the raw resolution of ERA-40 is 1.125deg and not 0.75deg (as ERA-Interim) suggesting that a first interpolation is made here. This could explain some differences between ERA40 and ERA-Interim over 1979-2001 and the problem of SST/SIC before and after 1979.

- line 11, pg 7: most of the observations used here were assimilated into the ERA-reanalysis which explains the good agreement (as also tells in the manuscript). It will be useful to add a comparison using the raw ERA outputs to check that the downscaling method does not bring additional uncertainties.

- line 12, pg 9: 5 to 25 mm / yr or month ? The units need to be more precise.

- line 10, pg 13: the 6 hourly outputs are linearly interpolated to 3min. A bilinear interpolation (allowing to represent the max/min of temperature) will be more adequate. Tmin, Tmax from ERA can be used to better represent the daily cycle with 6 hourly outputs. When the temperature is near 0deg, this could impact on the melt.

- Fig 6, pg 16 and line 6, pg 30: the SMB is mostly >0 in the North while is <0 in the South (y<8650km) everywhere. This is quite strange. Is it realistic? Do you have an explanation for this? For me, it seems rather to be an artefact from the use of the ERA (not representing these parts in its DEM) as forcing. This difference between south and north is less pronounced in MAR (Lang et al., 2015).

- Table 7, pg 23: the mean biases listed in Table 3 should be pointed on a map (e.g. on Fig 6).

- line 11, pg 25; line 12, pg 26, ...: the future projections presented in this paper are mostly very hazardous and speculative and are, for me, out of the scope of this paper which should focus only on present climate (as suggested in the title). It will be more robust to apply this methodology to GCM forcings and not ERA + anomalies for future projections.

- Sections 6.2 and 6.3, pg 27 should be put after Section 5.1.

---

## Referee Comment (RC2) · Anonymous Referee #2 · 12 Oct 2016

The authors propose a very high resolution reconstruction of the recent past and present climatic mass balance of glaciers in Svalbard. I agree with the remark of reviewer#1 on the conclusions of this paper but would like to add that the fact that this model allows the use of a higher spatial resolution than highly computationaly expensive physically based climate models is an advantage here since it is able to resolve the very alpine topography of Spitsbergen much better.

Downscaling precipitation is also not a straightforward task and the authors paid attention to the choice of their method.

The paper is generally well written but too much grammatical or spelling mistakes and typos remain. I have listed a series of corrections (see below) but a careful check by a native speaker would improve the readability of this manuscript a lot. Please also

check thoroughly the consistency of the units and the captions of the figures and tables (for example, you almost systematically leave a space after a left parenthesis in your tables).

I agree with the other remarks of reviewer#1 and mine are very minor changes or clarifications. I therefore encourage the publication of this paper.

P1, line 1: Longterm mass balance reconstruction/assessment

P1, line 13: 10-year period

P1, line 15: relatively to

P2, line 9: Radić

P3, lines 18 – 19: (Nordli et al., 2014; see also Supplement (Fig. S1-S2)).

P3, line 19: increase in

P3, line 20: have occurred

P3, line 26: in southern Spitsbergen

P3, line28: remove "in" before (see Fig. 1)

P3, line 30: retreating

p4, line 3: in central Spitsbergen

P4, line 11: slope and aspect were

P8, line 2: to constrain

P8, line 3: are likely incorporated

P8, line 8: please specify it is the Bayelva location since you already have another Ny-Alesund one.

P9, lines 1 – 4: the same sentence is repeated twice

P9, lines 9 – 10: which stations?

P10: caption of fig 5 in not clear. Are biases downscalled ERA – measured T ?

P10, line 7: provide

P10, line 10: usually has

P11, table 2: Tair bias is in °C whereas it is in K everywhere else

P11, line 3: 0.1, since . . .

P11, line 3: can only occur

P11, line 4: should anyway preclude

P11, line 4: as they are opaque

P11, line 4: thermal regions of the spectrum

P12, line 19: what does behaviour mean here? The special parametrization for thin layers? Or the fact that there is only a thin snow cover. Please modify the sentence

P12, line 23: following Gardner and Sharp (2010).

P12, line 25: SOMARS model

P12, line 27: are connected

P15, line 6: "accumulation area almost reaches sea level". Accumulation almost reach the coast or ELA reaches sea level would be a better formulation

P 15, line 10: -14 cm

P15, lines 9 – 14: you make it sound like you consider that the ERA-40 trend is unlikely to be real. Forcing your model with ERA-40 and ERA-Interim over their overlapping period as suggested by reviewer#1 and comparing the trends should also give an indication on wether the 1957 – 2014 trend is actually significant or if it is an artifact.

P15, line 16: melt season energy fluxes

P15, line 17: relatively instead of relative. And relatively to what?

P15 – 17: the use of a negative value for QM when there is melt is confusing in the whole section (also in the energy balance equation). Since QM is the result of the sum of all the other fluxes, it would make more sense to write the equation as QM = QNĂ+Ă... On p15, line 3 you talk about days with negative radiation balance and very little melt at the beginning of the melt season, which contradicts the sign of QM (negative when melt). On p16, line 3 + p17, line 1 you also write positive values for QM (13 and 30 W/m2). Finally, you should move the sentence about QM after QG since QM is the result and its value should be the conclusion of the paragraph. Also add that the resulting trend is significant.

P17, fig 7: y axis units: cm w.e.

P17, line 14: comprises

P18, line 1: amounts

P19, line 1: and, accordingly, the potential for heat release through refreezing is also reduced

P19, line 1 – 2: please rewrite the sentence starting with "These areas experience", it is not really clear what you mean.

P19, line 4: depend

P19, line 5: remove )

P21, lines 4-5: the model slightly underestimates

P21, line 8: remove also

P21, line 9: also shows

P21: the period you use for the validation is not clear. The right panels of fig. 11 and

the fact that you mention that there is no overlapping period with the Pinglot studies make it sound like you only use the period 2003 – 2014 like you do in your sensitivity experiments. Could you clarify that (also in the caption of fig. 11)?

P21, lines 18 – 20: you should carefully re-read and re-write these lines. E.g. To test the possible presence of a trend in the model performance, we compared the mass balance measured by stake at 380 m on Midtre Lovenbreen, a small glacier southeast of Ny-Alesund, to the modeled Bclim of a nearby pixel with the corresponding altitude. There is a good correlation . . .

P22, line 1: after 2000

P23, line 1: Due to large computational cost

P23, line 4: here we perturb

P23, line 8: relatively short

P23, line 10: The modeled Bclim

P25, line 2: without seasonality, by . . .

P25, line 3: results in

P25, lines 6 – 7: please re-write the sentence, it is not very clear at the first reading. E.g. However, the impact of this low sensitivity on Bclim is higher than for other glacierized regions given the low mass flux turnover in Svalbard.

P25, line 14: for (no capital F)

P25, line 26: the largest

P26, line 4: Tables 6 and 9

P26, line 5: shortwave

P26, line 6: indicative of

P26, lines 6 – 7: Last part of the sentence, what about temperature? Do you mean that albedo is also closely related to temperature since it controls the precipitation phase and the rate of albedo decay?

P26, line 13: As in Greenland . . .

P26, line 13: has been proposed

P26, line 15: the 1957-2014 period

P26, line 23: substantial amounts

P26, line 24: the remaining of winter

P26, line 28: in the near future

P26, line 34: mechanisms

P27, line 17: by Blaszczyk

P30, line 22: is not necessarily

P30, line 22: too low

P30, line 23: accumulates

P30, line 24: erodes

P30, line 25: indicate

P30, line 27: accumulation pattern

P30, line 31: by the elevation difference or by elevation differences

P30, line 34: considerably

P31, line 7: that have the highest

P31, line 23: reveals

P31, line 24: We assess

---

## Author Comment (AC1)

We are grateful for the efforts made by the two reviewers. Their comments and suggestions have been very helpful when revising the present MS. Regarding the major comments, we have assessed the heterogeneity effect of our composite forcing dataset by repeating simulations in the overlap period of ERA-40 and ERA-interim (1979-2002) using both re-analyses. We reply (black) to the individual comments (red) below and specify the associated revision of our MS (blue). In general, we have followed all advice and corrections to improve language and readability, and do not explicitly state these minor adjustments below.

**Referee #1**

This paper presents estimation of SMB at 1km of resolution over Svalbard downscaling directly the ERA reanalysis to 1km as in Hanna et al. (2008) over GrIS. This papers is generally well written and deserves to be published in TC although the main conclusions of this paper (lines 7-16, pg 32) just confirm previous results (e.g. Lang et al, 2015) and do not really bring new stuff. However, the methodology used is different than previous studies and gives results at 1km which will be very useful afterwards to force ice sheet models. Nevertheless, before publication, some issues need to be resolved if it is not a too big job for the authors.

As correctly noticed by R1, we use a different methodology that allows simulation of CMB at a considerably higher spatial resolution and/ or temporal coverage compared to previous studies of Svalbard glacier mass balance. In our MS, we comment on the importance of high spatial resolution to appropriately represent the hypsometric distribution of glaciers. Low spatial resolution causes an underestimation of glacierized area at low elevation, where highest mass loss rates are observed. An underestimation of this area introduces a bias into the mass balance estimate, as pointed out in our MS (P2L28). Furthermore, our study also covers the ERA40 period (1957-2002) and thus extends the simulation periods of previous, comparable assessments (Lang et al., 2015; Aas et al., 2016) by at least 2 decades. We regard the thorough evaluation of results and accompanying sensitivity tests, presented in our MS, as a major contribution to enhance reliability of RCM/ downscaling-based mass balance simulations. In our evaluation, we include a multitude of meteorological and glaciological datasets that have not been used in previous assessments. We therefore argue that our work considerably adds "new stuff".

- line 2, pg 32 and abstract: the authors suggests a change of SMB around 1980 which corresponds to the switch from ERA40 to ERAint. Discontinuities in SST/SIC are also mentioned around 1980 in the manuscript. How does this discontinuity impact on the results presented here? This change should be more discussed because it could be just an artefact of the use of inhomogeneous dataset. How do the results compare over 1980-2001 when both reanalysis are available? Estimation using ERA-40 over 1979-2001 should be added in this manuscript to check the homogeneity (assumed in the manuscript without proof: pg 31, line 3) of this SMB reconstruction over 1957-2014.

The discontinuity around 1980 is an important point that we discuss in our MS in the data section: P7 L15 to P8 L4, Results: P15 L 10-12, Discussion: P31 L5-6. To back up our argument, we have investigated the possibility that this discontinuity might have been caused by the heterogeneity of our composite forcing dataset that switches from ERA40 to ERAinterim in 1979. In doing so, we have re-run the model over the overlap period 1979-2002 using both re-analyses, but only for the grid points for which observations are available. We find that over 1979-2002, the ERA40-based simulation yields an about 13 cm w.e. higher mass balance than the ERAinterim-based one. This is caused by the generally lower summer temperatures in ERA40, slightly lower net radiation and lower wind speeds which

drive the turbulent fluxes. Regardless of this 13 cm w.e. difference, there is still a 20 cm drop of Bclim between 1970 and 1990 with only ERA40 as model forcing, see figure below. To investigate the possible impact of this heterogeneity, we also calculate Bclim trends over the period from 1958 to 2001 using three different data sets. 1) our original result with the combined ERA40-interim model forcing covering all of Svalbard. 2) Using the same ERA40/int forcing but only for the calibration sites (marked in Fig. 1). 3) Bclim at these locations only using the ERA40 forcing. Trends for these data sets are:

Bclim Trends over 1958-2001 +- 2 STD (cm w.e. per decade)
ERA40/int      All Svalbard:  -12.7 +- 5.7
ERA40/int      CalVal sites:  -11.2 +- 5.1
ERA40          CalVal sites:   -6.8 +- 3.9

[Figure]

*Figure shows annual area averaged Bclim of all Svalbard glaciers using the ERA40 (blue) and the ERAinterim (red). Based on the difference between the ERA40 and ERAinterim climate forcing at the CalVal sites for the overlap period we perturb the entire Svalbard record accordingly.*

Apparently, the switch from ERA-40 to ERA-interim in our composite forcing data, introduces a discontinuity. Nevertheless, the change in mass balance regime from predominantly positive to predominantly negative values around 1980 persists even if only ERA-40 is used as forcing until 2002. This suggests that this change in mass balance regime is not caused by the heterogeneity of our composite forcing. Nevertheless, we cannot rule out the possibility that this change was an artefact, caused by the discontinuity inherent in both reanalyses. It has been noted by several others that the inclusion of satellite data in the

assimilation from 1979 onwards represents a considerable quality increase of the re-analysis (Bromwich and Fogt, 2004; Screen and Simonds, 2011; Uppala et al., 2005).

We revised our MS as follows
P4 L27 (data section)

… and covers the periods 1957-2002 (ERA-40) and 1979-2014 (ERA-Interim).
To force our model, we select downscaled variables from ERA-40 for the period 1957-1978 and use ERA-interim from 1979 onwards. To investigate potential effects of this heterogeneity in our composite forcing, we have evaluated both datasets for the overlap period 1979-2002 at a number of points \ref{fig:DEM}.

In the discussion we added P 31 L6
Nevertheless, this discontinuity in re-analysis quality coincides with the discontinuity of our composite forcing dataset, that is based on ERA-40 before and on ERA-interim after 1979. To investigate the possibility that the resulting change in mass balance regime may be an artifact caused by this transition, simulations have been conducted over the overlap period 1979-2002 using both reanalyses, but only for the grid points used for calibration (\ref{fig_DEM}). We find that the ERA40-based simulation yields an about 13 cm w.e. higher mass balance than the ERAinterim-based one, but ERA-40 based simulations still show a 20 cm drop of Bclim between 1970 and 1990, larger than that caused by the dataset discontinuity. This suggests that this change in mass balance regime is not caused by the heterogeneity of our composite forcing. Nevertheless, we cannot rule out the possibility that this change was caused by the discontinuity inherent in both reanalyses due to availability of satellite observations after 1979. (Bromwich and Fogt, 2004; Screen and Simonds, 2011; Uppala et al., 2005).

Minor:
- line 26, pg 4: the raw resolution of ERA-40 is 1.125deg and not 0.75deg (as ERAInterim) suggesting that a first interpolation is made here. This could explain some differences between ERA40 and ERA-Interim over 1979-2001 and the problem of SST/SIC before and after 1979.
Our downscaling procedures for T, RH, WS, and radiation components basically represent sophisticated interpolations, hence interpolation of the underlying re-analysis should not have a noticeable impact. For the precipitation downscaling, area-averaged values for the driving variables are used to represent the advected air-mass. An interpolation of the underlying re-analysis should have no impact at all for this procedure. Our Bclim simulations do not directly account for SST/SIC but only uses the atmospheric variables of the underlying re-analysis which are influenced by SST/SIC. It is out of the scope of this study to further improve the quality of the reanalyses, instead we refine them to drive our Bclim model. In the discussion of our results, we try to distinguish between uncertainties due to our model and those inherent in the underlying re-analysis.

- line 11, pg 7: most of the observations used here were assimilated into the ERAreanalysis which explains the good agreement (as also tells in the manuscript). It will be useful to add a comparison using the raw ERA outputs to check that the downscaling method does not bring additional uncertainties.
We have added the unscaled ERA values in our new Tab 2. Since our downscaling first of all is a topographic correction, there is not much difference at the weather stations which are located around the coast. The exception is Holtedahlfonna which is situated several hundred meters above the coarse ERA-topography.

- line 12, pg 9: 5 to 25 mm / yr or month ? The units need to be more precise.
Per month, this is added in the text
- line 10, pg 13: the 6 hourly outputs are linearly interpolated to 3min. A bilinear interpolation (allowing to represent the max/min of temperature) will be more adequate. Tmin, Tmax from ERA can be used to better represent the daily cycle with 6 hourly outputs. When the temperature is near 0deg, this could impact on the melt.
Thanks for pointing this out, we have not been aware of the availability of Tmin and Tmax in the ERA dataset. This may be a valuable procedure to further refine the dataset in future applications. For the presented results, we do not have the possibility to repeat the entire simulation. Furthermore, we want to point out that our snowpack model uses the surface energy balance as Neumann condition at the snow surface, not just the surface temperature.

- Fig 6, pg 16 and line 6, pg 30: the SMB is mostly >0 in the North while is <0 in the South (y<8650km) everywhere. This is quite strange. Is it realistic? Do you have an explanation for this? For me, it seems rather to be an artefact from the use of the ERA (not representing these parts in its DEM) as forcing. This difference between south and north is less pronounced in MAR (Lang et al., 2015).
The mass balance gradient in NS direction has been found also by others (Aas et al., 2016) and qualitatively agrees with Fig 7a of Lang et al, 2015. Since our simulation better resolves the lower elevations, the lower minimum mass balance found in our results is expected. Nevertheless, we do not exclude the possibility that our downscaling scheme has some influence on this pattern and discuss this point on P27 L21 and on P30.

- Table 7, pg 23: the mean biases listed in Table 3 should be pointed on a map (e.g. on Fig 6).
Differences between modeled Bclim and ice core retrieved Bclim are indicated by colour-coded dots in the new Fig. 6

- line 11, pg 25; line 12, pg 26, ...: the future projections presented in this paper are mostly very hazardous and speculative and are, for me, out of the scope of this paper which should focus only on present climate (as suggested in the title). It will be more robust to apply this methodology to GCM forcings and not ERA + anomalies for future projections.
We agree that more reliable projections would require application of the entire model chain to GCM forcing, and this will be the subject of future work. We also agree that the outcome of our simplistic delta-approach is highly speculative, as we mention in the discussion (P26 L12). Therefore, we understand the applied scenarios as extended sensitivity tests to explore the realms of possibilities rather than as projections for future climate. Based on the results of our analysis and the conducted sensitivity tests, we discuss the controls on Bclim under changed conditions but we do not suggest a pathway for future evolution. This remains work to be done as outlined above.

- Sections 6.2 and 6.3, pg 27 should be put after Section 5.1.

The reviewer proposes to change the sequence in which results are presented and discussed, such that results and discussion would be ordered by topic, for instance Bclim results and associated uncertainties and context, then followed by energy balance results etc. Although this may be a valid alternative to present the material, we prefer to keep the original sequence, trying to strictly separate between description of results and their discussion.

**Referee #2**

**P8, line 8: please specify it is the Bayelva location since you already have another Ny-Alesund one.**
This was actually wrong in the manuscript. The radiation measurements are not from the Bayelva location, but from the BSRN-network located in Ny-Ålesund next to the weater station belonging to the Norwegian meteorological office. The two met-stations in Ny-Ålesund are only given one line in the table. The radiation measurements are credited in the text while the measurements from the Norwegian met-office are credited in the acknowledgment.
Changed to Ny-Ålesund in both text and Table

**P11, table 2: Tair bias is in ∘C whereas it is in K everywhere else**
Deg C is changed to K

**P12, line 19: what does behaviour mean here? The special parametrization for thin layers? Or the fact that there is only a thin snow cover. Please modify the sentence**
Precipitation events are frequent around Svalbard ($\sim$200 days a year), but usually yielding low amounts \citep{Aleksandrov-2005}.
This correction is essential to avoid that snow albedo is reset to fresh snow albedo in case of an insignificantly thin fresh snow layer.

**P15, line 6: "accumulation area almost reaches sea level". Accumulation almost reach the coast or ELA reaches sea level would be a better formulation**
ELA reaches sea level

**P15, line 17: relatively instead of relative. And relatively to what?**
compared to glaciers at lower latitudes

**P15 – 17: the use of a negative value for QM when there is melt is confusing in the whole section (also in the energy balance equation). Since QM is the result of the sum of all the other fluxes, it would make more sense to write the equation as QM = QNÂa+Âa...**
Regarding the sign convention for the surface energy balance we define a positive flux as an energy source at the surface.
Melt is an energy sink, hence negative.

**On p15, line 3 you talk about days with negative radiation balance and very little melt at the beginning of the melt season, which contradicts the sign of QM (negative when melt).**

**On p16, line 3 + p17, line 1 you also write positive values for QM (13 and 30 W/m2).**

Regarding the sign convention for the surface energy balance we define a positive flux as an energy source at the surface.
Melt is an energy sink, hence negative.

As pointed out, we have not been consistent, and numbers are changed to follow this sign convention.

Finally, you should move the sentence about QM after QG since QM is the result and its value should be the conclusion of the paragraph. Also add that the resulting trend is significant.

Sentences have been swapped and concerning the trend we have added:

The energy flux for melt increases over the study period by 3.8$\pm$1 \unit{W\,m^{-2}} per decade, meaning that the trend is significant despite large year-to-year variability.

P19, line 1 – 2: please rewrite the sentence starting with "These areas experience", it is not really clear what you mean.

several sentences are rewritten to clarify:

Henceforth, areas loosing firn due to raised ELA experience a cooling, which occurs at different altitudes around Svalbard similar to the regional ELA pattern.
In northeastern Svalbard cooling occurs above 100 \unit{m\,a.s.l.}, while cooling starts at 450 \unit{m\,a.s.l.} in southern Spitsbergen.
Figure~\ref{fig:REFT15m} shows the expansion of the cold ice area, which is predicted for polythermal glaciers in a warming climate, and also shown in model experiments \citep{Irvine-2011,Wilson-2013}.

P21: the period you use for the validation is not clear. The right panels of fig. 11 and the fact that you mention that there is no overlapping period with the Pinglot studies make it sound like you only use the period 2003 – 2014 like you do in your sensitivity experiments. Could you clarify that (also in the caption of fig. 11)?

Several modifications are done to clarify this aspect.
in section 2, the time period for mass balance measurements at the five glaciers are specified. When comparison between measured and modeled Bclim for all stake measurements, it is specified that this starts in 1987, when measurements begun at Kongsvegen.
Further, we emphasize which data (time and location) that are used in the calibration

In total, there are 1459 annual mass balances measurements covering different time periods at the various locations, see Section~\ref{sec:study} and Table~\ref{tab:IC} for specifications. Although mass balance stakes at these glaciers are in the proximity of the drilling sites, there is no overlap in time since retrieval of the ice cores coincides with the beginning of the stake measurements.
Mass balance measurements at Kongsvegen, Etonbreen and Hansbreen over 2004-2013 were used in the calibration and correspond to about 25 \% of the total measurements. Removing the measurements used for calibration, the independent dataset yields a slightly improved RMSE (58 \unit{cm\,w.e.\,yr^{-1}}), since we apply equal weights to all measurements..

Caption fig 11:
Note that the mass balance measurements at Kongsvegen, Etonbreen and Hansbreen over 2004-2013 also are used for model calibration

Regarding the lack of overlap, we specify that the ice cores are retrieved at the same time as the mass balance program began at the glacier of interest

P21, lines 18 – 20: you should carefully re-read and re-write these lines. E.g. To test the possible presence of a trend in the model performance, we compared the mass balance measured by stake at 380 m on Midtre Lovenbreen, a small glacier southeast of Ny-Alesund, to the modeled Bclim of a nearby pixel with the corresponding altitude. There is a good correlation . . .

We used your suggestion and also included the time period of mass balance measurements at Midtre Lovenbreen in the following sentence.

To test the possible presence of a trend in the model performance, we compared the mass balance measured by a stake at 380 m elevation on Midtre Lovenbreen, a small glacier southeast of Ny-Alesund, to the modeled Bclim of a nearby pixel with the corresponding altitude. There is a good correlation between the two records, although the model overestimates Bclim throughout the record from 1968-2014. However, after 2000 there is hardly any bias, while Bclim is overestimated by about 40 cmw.e. yr$^{-1}$ for the period 1968-1979. This trend in model performance is driven by melt, and therefore most likely by summer air temperatures. Although Bclim is overestimated, winter accumulation is underestimated in the model, which can be attributed to overestimation of sea ice in the reanalysis product at the northwest coast of Spitsbergen. The 5 west coast is usually ice-free year round, but prior to 1979 satellite data were not available to constrain sea ice cover and sea surface temperature in the reanalysis. An erroneous sea ice cover influence the heat and moisture uptake in the reanalyses, from which our forcing data were derived.

P26, lines 6 – 7: Last part of the sentence, what about temperature? Do you mean that albedo is also closely related to temperature since it controls the precipitation phase and the rate of albedo decay?

rewritten this part:
..., but is also correlated to temperature through the precipitation phase, rate of albedo decay and the importance of temperature for melt.

**References**

Bromwich, D. and R. Fogt, 2004: Strong Trends in the Skill of the ERA-40 and NCEP–NCAR Reanalyses in the High and Midlatitudes of the Southern Hemisphere, 1958–2001. *J. Climate,* **17**, 4603–4619, doi: 10.1175/3241.1.

Screen, J. and I. Simmonds, 2011: Erroneous Arctic Temperature Trends in the ERA-40 Reanalysis: A Closer Look. *J. Climate,* **24**, 2620–2627, doi: 10.1175/2010JCLI4054.1.

Uppala, S. M., KÅllberg, P. W., Simmons, A. J., Andrae, U., Bechtold, V. D. C., Fiorino, M., Gibson, J. K., Haseler, J., Hernandez, A., Kelly, G. A., Li, X., Onogi, K., Saarinen, S., Sokka, N., Allan, R. P., Andersson, E., Arpe, K., Balmaseda, M. A., Beljaars, A. C. M., Berg, L. V. D., Bidlot, J., Bormann, N., Caires, S., Chevallier, F., Dethof, A., Dragosavac, M., Fisher, M., Fuentes, M., Hagemann, S., Hólm, E., Hoskins, B. J., Isaksen, L., Janssen, P. A. E. M., Jenne, R., Mcnally, A. P., Mahfouf, J.-F., Morcrette, J.-J., Rayner, N. A., Saunders, R. W.,

Simon, P., Sterl, A., Trenberth, K. E., Untch, A., Vasiljevic, D., Viterbo, P. and Woollen, J. (2005), The ERA-40 re-analysis. Q.J.R. Meteorol. Soc., 131: 2961–3012. doi:10.1256/qj.04.176